# Multiple cancer pathways regulate telomere protection

Leire Bejarano[1,†], Giuseppe Bosso[1,†], Jessica Louzame[1], Rosa Serrano[1], Elena Gómez-Casero[2], Jorge Martínez-Torrecuadrada[3], Sonia Martínez[2] (ID), Carmen Blanco-Aparicio[2], Joaquín Pastor[2] & Maria A Blasco[1,*] (ID)

## Abstract

Telomeres are considered as universal anti-cancer targets, as telomere maintenance is essential to sustain indefinite cancer growth. Mutations in telomerase, the enzyme that maintains telomeres, are among the most frequently found in cancer. In addition, mutations in components of the telomere protective complex, or shelterin, are also found in familial and sporadic cancers. Most efforts to target telomeres have focused in telomerase inhibition; however, recent studies suggest that direct targeting of the shelterin complex could represent a more effective strategy. In particular, we recently showed that genetic deletion of the TRF1 essential shelterin protein impairs tumor growth in aggressive lung cancer and glioblastoma (GBM) mouse models by direct induction of telomere damage independently of telomere length. Here, we screen for TRF1 inhibitory drugs using a collection of FDA-approved drugs and drugs in clinical trials, which cover the majority of pathways included in the Reactome database. Among other targets, we find that inhibition of several kinases of the Ras pathway, including ERK and MEK, recapitulates the effects of *Trf1* genetic deletion, including induction of telomeric DNA damage, telomere fragility, and inhibition of cancer stemness. We further show that both bRAF and ERK2 kinases phosphorylate TRF1 *in vitro* and that these modifications are essential for TRF1 location to telomeres *in vivo*. Finally, we use these new TRF1 regulatory pathways as the basis to discover novel drug combinations based on TRF1 inhibition, with the goal of effectively blocking potential resistance to individual drugs in patient-derived glioblastoma xenograft models.

**Keywords** drug resistance; ERK kinase; glioblastoma; telomeres; TRF1 inhibitors

**Subject Categories** Cancer; Pharmacology & Drug Discovery

See also: **J Cherfils-Vicini & E Gilson** (July 2019)

## Introduction

Telomeres are heterochromatic structures at the ends of chromosomes, which are essential for chromosome stability (Blackburn, 1991). They are composed by tandem repeats of the TTAGGG repetitive sequence bound by the so-called shelterin complex, which is formed by six proteins named TRF1, TRF2, RAP1, TPP1, TIN2, and POT1. The shelterin complex constitutes the so-called capping of the telomeres, which is essential for their protection, preventing telomeres from fusion to other chromosome ends, from telomere fragility, and from degradation (De Lange, 2005). Also, the shelterin complex prevents the recognition of telomeres as double-strand DNA breaks (DSB) and the subsequent activation of a persistent DNA damage response (DDR; De Lange, 2005). Telomeres get shorter with aging as cells divide to regenerate tissues, and when they reach a very short length, they can contribute to physiological aging (Hemann & Greider, 2000; Samper *et al*, 2001). Telomere shortening can be compensated through *de novo* addition of telomeric repeats by telomerase, a reverse transcriptase composed by a catalytic subunit (TERT) and an RNA component (Terc; Greider & Blackburn, 1985). Telomeres can also be elongated by an alternative mechanism known as alternative lengthening of telomeres (ALT), which is based in homologous recombination between chromosome ends (Bryan *et al*, 1997).

The majority of cancer cells aberrantly activate telomerase or ALT mechanisms to be able to divide indefinitely (Kim *et al*, 1994; Bryan *et al*, 1995; Shay & Bacchetti, 1997; Barthel *et al*, 2017). More than 90% of human tumors aberrantly overexpress telomerase (Kim *et al*, 1994; Shay & Bacchetti, 1997; Joseph *et al*, 2010), while the remaining telomerase-negative tumors activate ALT (Bryan *et al*, 1997; Barthel *et al*, 2017). For this reason, telomeres have been considered as potential anti-cancer targets.

Most studies have focused in telomerase inhibition as therapeutic approach for telomere targeting in cancer. One of the most advanced anti-telomerase therapies is GRN163L, also called Imetelstat (Harley, 2008; Joseph *et al*, 2010). However, clinical trials for several cancer types had shown that this strategy has some limitations (Parkhurst

1  Telomeres and Telomerase Group, Molecular Oncology Program, Spanish National Cancer Centre (CNIO), Madrid, Spain
2  Experimental Therapeutics Program, Spanish National Cancer Centre (CNIO), Madrid, Spain
3  Biotechnology Program, Spanish National Cancer Centre (CNIO), Madrid, Spain
   *Corresponding author. Tel: +34 91 732 8031; Fax: +34 91 732 8028; E-mail: mblasco@cnio.es
   †These authors contributed equally to this work

et al, 2004; Middleton et al, 2014; El Fassi, 2015). In mouse models of cancer, the anti-tumorigenic effect of telomerase inhibition is only achieved when telomeres reach a critically short length, and this effect is lost in the absence of the p53 tumor suppressor gene, which is frequently mutated in cancer (Gonzalez-Suarez et al, 2000; Perera et al, 2008). Also, cancer cells could activate ALT to overcome telomerase inhibition. Thus, alternative therapies of telomere targeting should be developed in order to efficiently target telomeres in cancer.

In this regard, our group and others have found that not only telomerase but also shelterin proteins are often mutated in cancer. In particular, POT1 is mutated in several types of sporadic and familial human tumors, including chronic lymphocytic leukemia (CLL; Ramsay et al, 2013), familial melanoma (Robles-Espinoza et al, 2014; Shi et al, 2014), Li–Fraumeni-like families (LFL) with cardiac angiosarcomas (CAS; Calvete et al, 2015), glioma (Bain-bridge et al, 2015), mantle cell lymphoma (Zhang et al, 2014), and parathyroid adenoma (Newey et al, 2012). Also, previous studies from our group suggested that targeting the shelterin complex through inhibition of one of its central components, TRF1, leads to a rapid telomere dysfunction independently of telomere length, thus avoiding the shortcomings of telomerase inhibition (Garcia-Beccaria et al, 2015; Bejarano et al, 2017). In particular, we reported that induction of telomere uncapping by either Trf1 genetic depletion or TRF1 chemical inhibition can effectively block initiation and progression of aggressive tumors in both lung cancer and glioblas-toma mouse models, in a manner that is independent of telomere length (Garcia-Beccaria et al, 2015; Bejarano et al, 2017). We further demonstrated that TRF1 abrogation in normal tissues was tolerated and did not result in decreased mouse survival or severe defects in tissues (i.e., TRF1 abrogation did not affect brain olfactory or memory functions nor affected highly proliferative tissues) (Gar-cia-Beccaria et al, 2015; Bejarano et al, 2017). These findings were recapitulated by using TRF1 chemical inhibitors. In particular, we found that TRF1 is phosphorylated at different residues by AKT and that these modifications regulate TRF1 foci formation in vivo (Men-dez-Pertuz et al, 2017). Thus, PI3K inhibitors, as well as inhibitors of the PI3K downstream target AKT, significantly reduced TRF1 telomeric foci and lead to increased telomeric DNA damage and fragility, also impairing the growth of lung and GBM cells (Garcia-Beccaria et al, 2015; Bejarano et al, 2017).

Given these results, here we set to discover additional TRF1 modulatory pathways by carrying out a screening with FDA-approved drugs or drugs that are currently in clinical trials, and which cover the majority of known cancer pathways. We found several drugs that inhibit TRF1 independently of the Pi3K pathway, including inhibitors of some of the most deregulated pathways in cancer. Among these pathways, here we demonstrate an unprece-dented role of the Ras pathway in regulating telomere protection. Finally, we used these new TRF1 regulatory pathways as a rational to discover novel drug combinations based on TRF1 inhibition, which effectively block resistance to individual drugs in patient-derived glioblastoma mouse models. The results shown here uncover the importance of telomere capping for cancer cells and identify novel therapeutic strategies based on telomere targeting.

## Results

### Identification of novel TRF1 regulatory pathways

To identify novel pathways that regulate TRF1 protein levels, we used a cell-based high-throughput screening to determine TRF1 foci fluorescence upon treatment with different drugs (Garcia-Beccaria et al, 2015). We screened a CNIO collection of 114 anti-tumoral drugs, which are either approved by the Food and Drug Administra-tion (FDA) or in clinical trials, and which cover 20 of the 26 path-ways included in Reactome database (Fig EV1A). To this end, we treated CHA-9.3 mouse lung cancer cells (Garcia-Beccaria et al, 2015) with the compounds at 1 μM concentration during 24 h, followed by immunofluorescence analysis using anti-TRF1 antibody to quantify TRF1 foci fluorescence (Fig 1A). As positive controls, we used PI3K inhibitors (PI3Ki), previously shown by us to inhibit TRF1 telomeric foci (Mendez-Pertuz et al, 2017).

We found several drugs with the ability to downregulate TRF1 levels, including inhibitors of some of the most deregulated path-ways in cancer. In particular, we identified inhibitors of the Ras pathway, including ERK and MEK inhibitors (ERKi and MEKi); compounds related with the cell cycle, such as Aurora inhibitors (Aurorai), CDK inhibitors (CDKi), and PLK1 inhibitors (PLK1i); an inhibitor of the chaperone HSP90 (HSP90i); two chemotherapeutic agents named gemcitabine and docetaxel; and, as expected, several compounds related with the PI3K pathway, including mTOR inhibitors (mTORi) and RTK inhibitors (RTKi) (Fig 1B; see Appendix Table S1 for results with all drugs tested). To test whether the inhibitors found to decrease TRF1 levels also decreased the levels of other shelterin components, we studied RAP1 and TIN2 protein levels (Appendix Fig S1A and B). We found that TIN2 protein levels were not affected by any of the compounds found to decrease TRF1, with the exception of RTKi that increased TIN2

**Figure 1. Identification of novel compounds with the ability to downregulate TRF1 protein levels.**

A Experimental procedure: 114 compounds approved by the FDA or in clinical trials are assessed by the Opera High Screening system for their ability to reduce TRF1 protein levels in CHA9.3 lung cancer mouse cells.

B Representative images (top) and quantification (bottom) of TRF1 nuclear fluorescence of patient-derived h676 GSCs cells treated with the indicated compounds for 24 h at 1 μM. Scale bars, 5 μm. Data are representative of n = 3 biological replicates.

C Western blot images (top) and TRF1 protein levels (bottom) of patient-derived h676 GSCs cells treated with the indicated compounds for 24 h at 1 μM. Data are representative of n = 3 (PLKi, HSP90i, and RTKi) and n = 4 (Aurorai, mTOR, CDKi, docetaxel, gemcitabine, ERKi, MEKi) biological replicates.

D Schematic representation of the novel TRF1 regulatory pathways. Asterisk indicates targets of TRF1 inhibitory compounds found in the screening.

Data information: Data are represented as mean ± SEM. n represents biological replicates. Significant differences using unpaired t-test are indicated by *P < 0.05, **P < 0.01, ***P < 0.001.

Source data are available online for this figure.

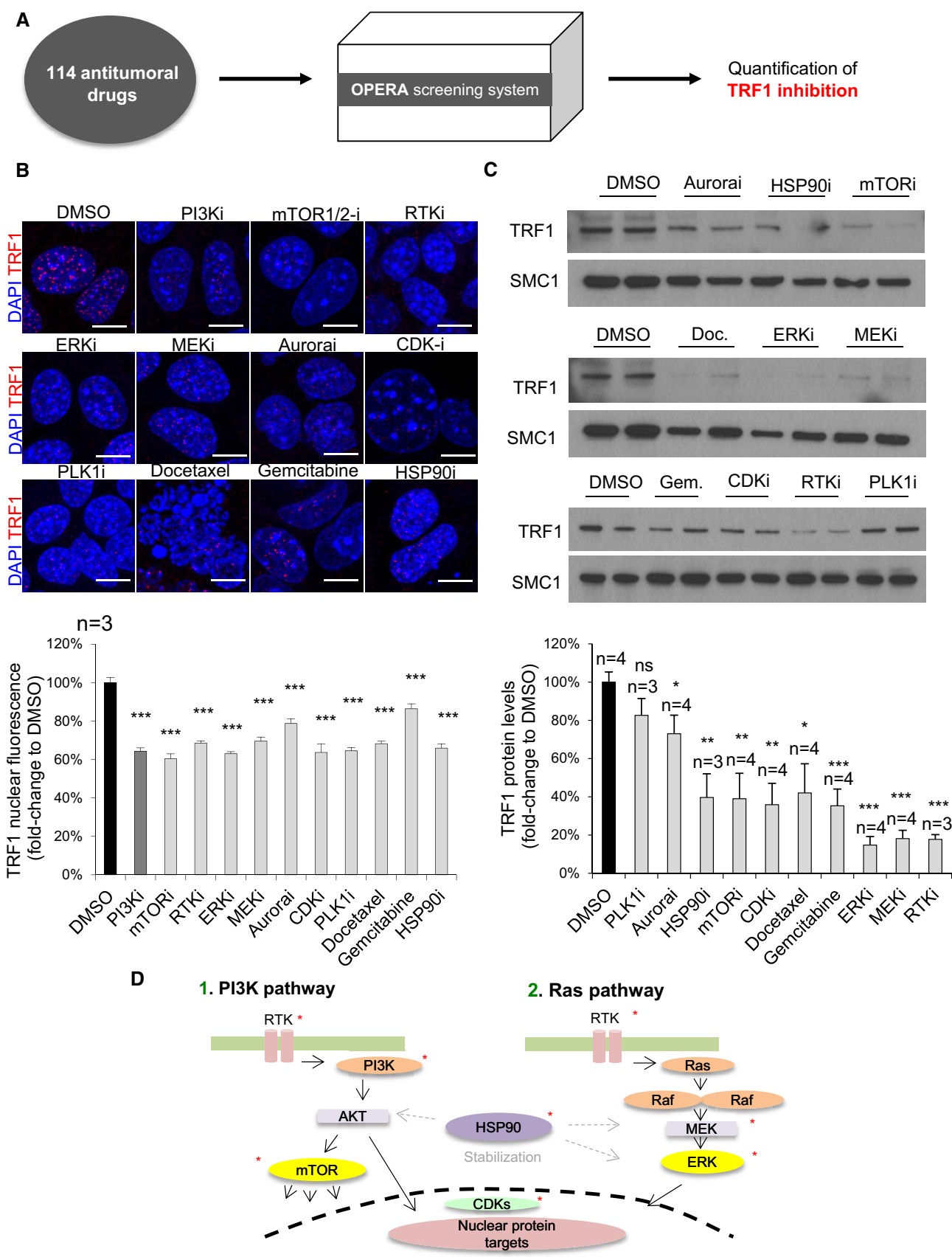

**Figure 1.**

levels (Appendix Fig S1A). In the case of RAP1, we found a significant decrease by MEKi, again docetaxel increased RAP1 protein levels (Appendix Fig S1B).

To validate the ERK/MEK pathway as a novel signaling pathway that modulates TRF1 foci formation, we selected 5 structurally different MEK inhibitors and three structurally different ERK inhibitors (see Materials and Methods) and tested these compounds at 1 μM during 24 h in the CHA-9.3 mouse lung cancer cell line followed by TRF1 immunofluorescence analysis. All the MEK and ERK inhibitors showed a clear inhibition of TRF1 levels (Fig EV1B and C) with the exception of the ERK inhibitor GDC-0944, which, in fact, did not inhibit phospho-ERK levels in this cell line, explaining the lack of effect on TRF1 foci (Fig EV1D). The same chemical biology approach was used to validate HSP90 and tubulin agents as TRF1 modulators. We tested a total of five structurally different HSP90 inhibitors and three independent taxol derivatives (Materials and Methods). All of the compounds rendered a significant decrease in TRF1 foci fluorescence (Fig EV1E and F). In summary, we identify here new TRF1 regulatory pathways and validate these novel TRF1 modulators by using several chemically diverse inhibitors of these pathways.

As we previously described that TRF1 inhibition effectively blocked glioblastoma growth both in mouse models and in xenograft models of patient-derived glioma stem cells (GSCs; Bejarano et al, 2017), we next tested whether these newly identified TRF1 modulators were also able to downregulate TRF1 protein levels in patient-derived GSCs. To this end, we treated h676 patient-derived primary GSCs (Bejarano et al, 2017) with the different compounds at 1 μM concentration during 24 h and analyzed TRF1 levels by Western blot. All the compounds, with the exception of Aurora and PLK1 inhibitors, were able to downregulate TRF1 protein levels in human GSCs (Fig 1C); thus, we further characterized all the compounds except for Aurora and PLK1 inhibitors.

In summary, by using an unbiased screening we identified here several pathways and compounds with the ability to downregulate TRF1 levels in both lung cancer and glioblastoma cells, including inhibitors of the Ras pathway (ERK and MEK), the cell cycle-related CDK inhibitor, the inhibitor of the chaperone HSP90, RTK, and mTOR inhibitors (Fig 1D), and two chemotherapeutic agents (gemcitabine and docetaxel).

## Novel TRF1 modulators induce telomeric DNA damage in cancer cells

*Trf1* deletion has been previously shown to induce a persistent DDR at telomeres in different cell lines, which leads to decreased cell viability (Martinez *et al*, 2009; Sfeir *et al*, 2009). To address whether the newly identified TRF1 inhibitors were also able to induce DNA damage specifically at telomeres (the so-called telomere-induced DNA damage foci or TIFs), we treated CHA9.3 lung cancer cells with all the selected hits for 24 h at 1 μM followed by a double immunofluorescence of the γH2AX to detect DNA damage and of the shelterin component RAP1 to mark the telomeres. We found that all the TRF1 compounds were able to significantly increase the number of cells with more than 1 TIF, with the exception of mTOR inhibitors, where the TRF1 decreased did not reach statistical significance (Fig 2A). Furthermore, when we normalized the mean number of TIFs per nucleus (Appendix Fig S2A) to total γH2AX DNA damage (Appendix Fig S2B), we observed that in the majority of the cases the DNA damage steams from telomeres (Appendix Fig S2C). Importantly, in patient-derived glioblastoma stem cells (h676 GSCs), we found that all the TRF1 inhibitors induced increased global DNA damage as indicated by increased numbers of cells positive for the γH2AX DNA damage marker (Fig 2B); however, in this case owing to the fact that these cells cannot be attached to the plates, we could not perform TIF analysis.

*Trf1* inhibition by using genetic deletion has been previously shown to induce the so-called multitelomeric signals (MTS), which are associated with increased telomere fragility and increased telomere damage (Martinez *et al*, 2009; Sfeir *et al*, 2009). Thus, we next tested whether our hits had the ability to increase the number of MTS in h676 patient-derived GSCs. To this end, we treated GSCs with the different compounds for 24 h at 1 μM and then performed telomere quantitative fluorescence in situ hybridization (Q-FISH) on metaphase spreads to visualize telomeres (Materials and Methods). We found that mTOR inhibitors, MEK inhibitors, and RTK inhibitors significantly increase the number of MTS (Fig 2C). The rest of the compounds, namely ERK inhibitors, CDK inhibitors, HSP90 inhibitors, docetaxel, and gemcitabine, completely blocked the formation of metaphases, and this prevented determination of MTS upon treatment with these compounds. In summary, the newly identified TRF1 inhibitory compounds recapitulate the telomeric defects associated with TRF1 genetic deletion, such as induction of telomeric DNA damage (TIFs) and induction of telomere fragility (MTS).

## Novel TRF1 inhibitors reduce stemness of primary patient-derived GSCs

We previously demonstrated that TRF1 is overexpressed in both adult and pluripotent stem cells and it is essential for stemness (Schneider *et al*, 2013) and that *Trf1* genetic deletion significantly reduced stemness in both neural stem cells (NSCs) and glioma stem

---

**Figure 2.   New TRF1 chemical inhibitors induce DNA damage in lung cancer and glioblastoma cells.**

A   Representative images (left) and percentage (right) of cells presenting 1 or more γH2AX and RAP1 colocalizing foci (TIFs) upon treatment of CHA9-3 lung cancer cells with the indicated compounds. White arrowheads point to colocalization of γH2AX and RAP1. Scale bars, 5 μm. Data are representative of n = 6 (DMSO) and n = 3 (mTORi, PI3Ki, RTKi, MEKi, ERKi, HSPO90i, CDKi, docetaxel) biological replicates.

B   Representative images (left) and percentage (right) of γH2AX-positive cells per field in DMSO or compound-treated patient-derived h676 GSCs. Scale bars, 50 μm. Data are representative of 6 (DMSO) and 3 (mTORi, docetaxel, ERKi, MEKi, RTKi, HSP90i, gemcitabine, CDKi) biological replicates.

C   Quantification of multitelomeric signals (MTS) in patient-derived h676 GSC metaphases upon treatment with the indicated compounds. Representative images of the qFISH in the metaphases (left). Multitelomeric signals are indicated by arrowheads. Scale bars, 1 μm. Data are representative of n = 31 (DMSO), n = 18 (mTORi), n = 11 (MEKi), and n = 24 (RTKi) biological replicates.

Data information: Data are represented as mean ± SEM. Significant differences using unpaired *t*-test are indicated by *P < 0.05, **P < 0.01, ***P < 0.001.

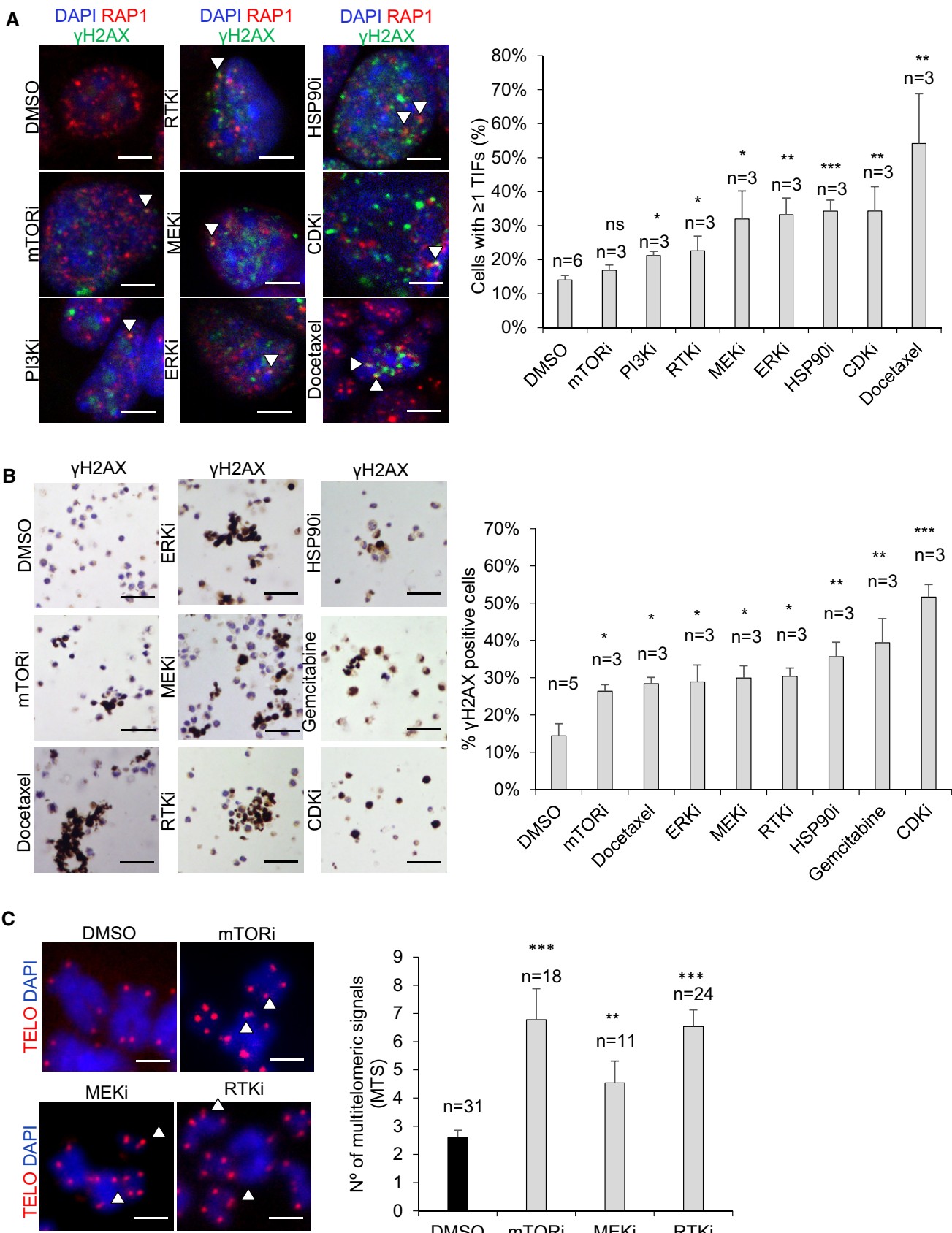

**Figure 2.**

cells (GSCs; Bejarano et al, 2017). Thus, we next set to address whether the novel TRF1 modulators identified here also have the ability to reduce stemness in two independent patient-derived GSCs, h676, and h543 cells (Bejarano et al, 2017). To this end, we performed a dose–response sphere formation assay by plating disaggregated cells in a 96-well plate and treated them with the different compounds at several concentrations (Materials and Methods). We found that all the compounds had the ability to reduce stemness in both patient-derived GSC cell lines as indicated by decreased number of spheres 7 days after plating (Fig 3A–H). It is important to point out that in these graphs, the data are represented as a percentage of spheres normalized to the DMSO cells. In addition, we calculated the growth inhibition 50 (or GI50) for every compound, which indicates the concentration at which the drug causes the 50% of reduction in stemness (Table 1). We observed that the most potent compounds were the HSP90 inhibitor, gemcitabine and docetaxel, which had a GI50 ranging between 0.1 and 1.5 nM. They were followed by mTOR inhibitor, ERK inhibitor, MEK inhibitor, and ERK inhibitor in which the GI50 was between 0.05 and 0.5 μM, and finally, the RTK inhibitors showed the highest GI50, ranging between 1 and 5 μM (Table 1). The fact that the RTK inhibitor, which strongly decreased TRF1 protein levels 24 h after treatment, had a high GI50 may be due to a low stability of this compound in long-lasting experiments. In summary, all the newly identified TRF1 modulators significantly reduce the stemness of two independent primary patient-derived GSCs lines.

### TRF1 is directly phosphorylated by ERK2, mTOR, and bRaf kinases

Next, we set to study the mechanisms by which the newly identified compounds regulate TRF1 levels. We had previously reported that the TRF1 telomeric protein is regulated by the PI3K signaling pathway (Mendez-Pertuz et al, 2017). In particular, TRF1 is directly phosphorylated by the PI3K downstream target AKT at different residues (T248, T330, and S344), and this phosphorylation is necessary for TRF1 stability and TRF1 foci formation in vivo (Mendez-Pertuz et al, 2017). Thus, we first addressed here whether the newly identified compounds were acting through the PI3K/AKT pathway by assessing p-AKT levels upon treatment with the different inhibitors. To this end, we treated h676 GSCs with the compounds at 1 μM for 24 h and we checked p-AKT levels by Western blot. From all the tested compounds, RTKi, mTORi, and HSP90i significantly downregulated AKT phosphorylation (Appendix Fig S3A). However, we observed that ERKi, MEKi, CDKi, gemcitabine, and docetaxel did not affect p-AKT, suggesting that these compounds are acting through AKT-independent pathways (Appendix Fig S3A).

As the Ras pathway is heavily mutated in cancer, we next set to explore whether TRF1 is also a direct substrate of different kinases of the Ras pathway—including ERK, MEK, and bRaf kinases. In parallel, we also tested whether TRF1 was a target of the mTOR kinase, also found here to modulate TRF1 (Fig 1A and B). To this end, we carried out in vitro kinase assays with affinity-purified mouse GST-TRF1 incubated with either mouse-purified ERK2, mouse-purified MEK1, human-purified bRaf, or human-purified mTOR, always in the presence of [γ-$^{32}$P]ATP (Materials and Methods). Importantly, ERK2 and bRaf but not MEK yielded a clear TRF1 phosphorylation signal (Fig 4A–D). Interestingly, an oncogenic mutant of bRaf (V600E; Davies et al, 2002) triggered a significantly

higher TRF1 phosphorylation compared to the wild-type bRaf kinase (Fig 4C), thus suggesting a potentially important role of the bRaf kinase in TRF1 regulation in cancer.

Next, we studied the specificity of these phosphorylation signals by using specific inhibitors of these kinases. We observed that TRF1 phosphorylation by ERK2 was decreased in the presence of ERK inhibitors but not MEK inhibitors (Fig 4E). Similarly, TRF1 phosphorylation by bRaf was inhibited in the presence of the bRaf inhibitors vemurafenib and dabrafenib (Fig 4F). Regarding mTOR kinase, we observed a TRF1 phosphorylation signal which was decreased in the presence of the TRF1 inhibitor Ku007694 but not in the presence of rapamycin, in agreement with the fact that rapamycin acts through FKBP12 (Fig 4G and H).

We next set to identify the specific TRF1 phosphorylation sites by ERK2, bRaf (WT and V600E), and mTOR kinases. To this end, we analyzed phosphopeptides by liquid chromatography–mass spectrometry (LC/MS/MS) analysis in TRF1, TRF1&ERK2, TRF1&bRAF$^{WT}$, TRF1&bRAF$^{V600E}$, and TRF1&mTOR samples containing ATP (Materials and Methods). Interestingly, we identified 13 different TRF1 phosphopeptides in samples containing ERK2, including T4, S6, S7, T44, T195, T268, T270, T274, T298, T328, T330, T335, and T358 (Fig 4I). Additionally, we identified four different TRF1 phosphopeptides in samples containing RAF$^{WT}$ and bRAF$^{V600E}$, including T4, T298, T330, and T336 (Fig 4J). Interestingly, the signal of the phosphopeptides was higher in the presence of the mutant form of bRAF (bRAF$^{V600E}$) (Fig 4J), again suggesting a potentially important role of TRF1 modification by bRAF in cancer. In the case of the mTOR kinase, we identified 13 different residues, including T4, S6, S7, T44, S176, S236, T241, T268, S301, S319, T330, S352 (Fig 4K). As expected, no phosphopeptides were identified in the sample containing only TRF1 (Fig 4I–K). Out of the identified phosphorylation sites (Fig 4L), we decided to focus in the ERK2-dependent phosphosites for further analysis as this kinase is downstream in the Ras pathway and is also AKT independent (Appendix Fig S3A). Future studies warrant further analysis on the role of mTOR and bRaf kinases in TRF1 regulation.

For the in vitro validation of the ERK phosphorylation sites, we generated the GST-tagged Trf1 alleles T44, T195, T298, and T358 as singles mutants and T4/S6/S7, T268/T270/T274, and T328/T330/T335 as triple mutants. In all the cases, threonine or serine was mutated to alanine. The affinity-purified GST-TRF1 WT or mutant alleles were incubated with mouse-purified ERK2 always in the presence of [γ-$^{32}$P]ATP. We found significantly decreased TRF1 phosphorylation levels in the variants harboring T4/S6/S7, T44, T268/T270/T274, and T328/T330/T335 substitutions compared to wild-type TRF1 (Fig 4M). We extended this analysis with additional TRF1 single mutants in ERK-phosphorylation sites, such as T328A, T330A, and T335A (Fig 4N), all of which resulted in decreased ERK-dependent TRF1 phosphorylation. Furthermore, we demonstrate that, among the AKT-dependent phosphosites of TRF1, S344 (T358 in human) is as also a target for ERK-mediated phosphorylation (Fig 4O). As negative control, we also generated a TRF1 phosphomutant in residue T248 whose phosphorylation is mediated by AKT but not ERK (Fig 4O; Mendez-Pertuz et al, 2017). As expected, this mutant did not result in decreased TRF1 phosphorylation (Fig 4O).

Importantly, in order to address the in vivo role of TRF1 modifications by ERK2, eGFP-tagged Trf1 wild-type and mutant alleles

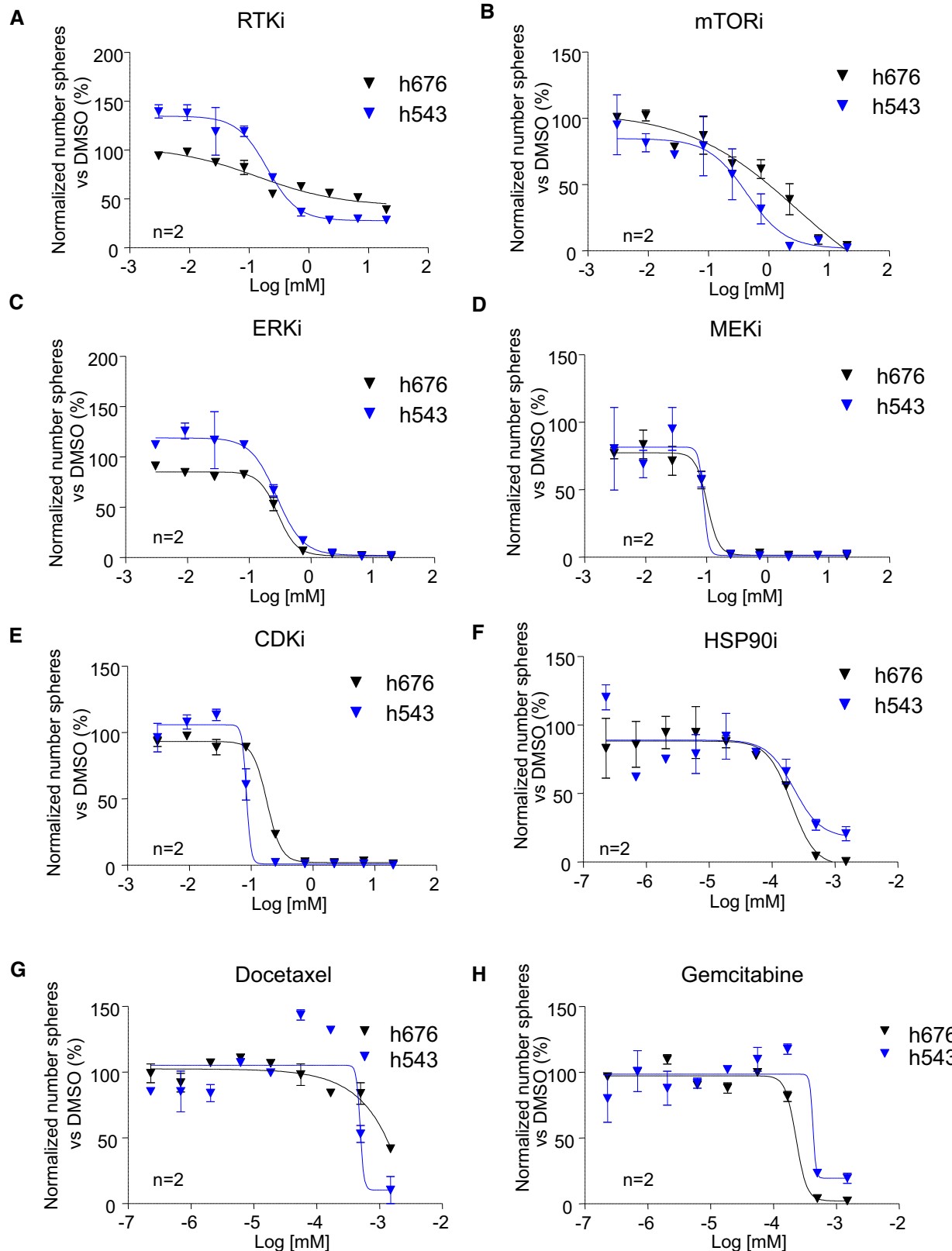

**Figure 3. New TRF1 inhibitory compounds reduce stemness in patient-derived glioma stem cells.**

A–H Dose–response curves of h543 and h676 patient-derived GSCs treated with the indicated compounds at several concentrations. Data are representative of $n = 2$ biological replicates. Data are represented as mean ± SEM normalized to DMSO.

**Table 1. Growth inhibition 50 of the selected compounds in h676 and h543 GSCs.**

| Compound | CNIO code | GI50 (nM) h676 | GI50 (nM) h543 |
|---|---|---|---|
| HSP90i (Geldanamycin) | ETP-50853 | 0.15 | 0.25 |
| Docetaxel | ETP-45335 | 0.24 | 0.46 |
| Gemcitabine | ETP-45337 | 1.25 | 0.5 |
| mTORi | ETP-50537 | 433 | 125 |
| ERKi | ETP-50728 | 218 | 354 |
| MEKi | ETP-51677 | 61 | 90 |
| CDKi | ETP-47306 | 164 | 86 |
| RTKi (Dasatinib) | ETP-51801 | 4,400 | 913 |

were transduced into p53-deficient $Trf1^{lox/lox}$ MEFs (Fig 5A) and we determined nuclear eGFP spot fluorescence, as an indication of the ability of the different eGFP-TRF1 mutant proteins to localize to telomeres. From the 13 possible phosphosites, T44, T195, T298, and T358 were generated as single mutants, whereas T4/S6/S7, T268/T270/T274, and T328/T330/T335 were assessed as triple mutants (Fig 5A). In order to discard possible interference of the eGFP-tagged TRF1 alleles with the endogenous TRF1, MEFs were transduced with Cre recombinase to induce endogenous $Trf1$ deletion. Overexpression of eGFP-$Trf1$ alleles and endogenous $Trf1$ deletion were confirmed by Western blot analysis using a specific TRF1 antibody (Fig 5B). Quantification of nuclear eGFP spot fluorescence in $Trf1^{-/-}$ MEFs showed that the triple mutant T328/T330/T335 was the only mutant which showed a significant decrease in the intensity of TRF1 telomeric foci compared to MEFs expressing wild-type TRF1 (Fig 5C). In contrast, no significant differences were detected between the rest of the mutants (T44, T195, T298, T358, T4/S6/S7, T268/T270/T274) and wild-type TRF1 (Fig 5C; see representative images in Fig EV2). TRF1 deficiency is associated with severe defects in cell proliferation (Martinez *et al*, 2009; Sfeir *et al*, 2009). In order to address *in vivo* whether the different TRF1 mutants were able to rescue the proliferation defects of $Trf1$-deficient MEFs, we assess the growth rate of $Trf1$-deficient MEFs expressing the different eGFP-tagged $Trf1$ wild-type or mutant alleles. All the single mutants were able to completely or almost completely rescue the proliferation defects associated with $Trf1$ deficiency (Fig 5D). We

next assessed the triple mutants (T4/S6/S7, T268/T270/T274, and T328/T330/T335), and, in agreement with eGFP-TRF1 telomeric foci findings (Fig 5C), we observed that the triple mutant TRF1-T328/T330/T335 showed the more severe proliferation defects (Fig 5D). Thus, we decided to study the effect of each of the single mutants TRF1-T328, TRF1-T330, and TRF1-T335 separately by transducing the mutant alleles into $Trf1^{lox/lox}$ MEFs, followed by Cre recombinase transduction to induce endogenous $Trf1$ deletion. Overexpression of eGFP-$Trf1$ alleles and endogenous $Trf1$ deletion were confirmed by Western blot analysis using a specific TRF1 antibody (Fig 5E). Quantification of eGFP-TRF1 nuclear fluorescence revealed that the mutant TRF1-T330 and, in a lesser extent, the mutant TRF1-T335 showed a decrease in the intensity of TRF1 telomeric foci compared to MEFs expressing wild-type TRF1 (Fig 5F; see representative images in Fig EV2). Interestingly, out of the three mutants, the TRF1-T330 showed the more severe proliferation defects (Fig 5D). Next, we plotted the proliferation results in parallel and calculated the statistical significance for the different eGFP-TRF1 proteins (Appendix Fig S4). Albeit all the mutants showed a partial rescue of the proliferation defects, mutants T330A and T335A resulted in a significantly lower proliferation rate compared with wild-type GFP1-TRF1 (Appendix Fig S4). It is important to point out that T330 phosphosite is also essential for AKT-mediated TRF1 phosphorylation (Mendez-Pertuz *et al*, 2017).

Importantly, we next set to address whether we could rescue the telomere damage phenotype induced by ERKi by overexpressing wild-type eGFP-TRF1 or different TRF1 mutants. To this end, we overexpressed WT TRF1-eGFP and non-phosphorylatable mutant versions of TRF1 in MEFs treated or not with the ERKi (Fig 5G). In all cases, treatment with the ERKi decreased the levels of wild-type or mutant eGFP-TRF1 proteins (Fig 5G). Interestingly, the induction of TIFs as the result of ERK1/2 inhibition can be reduced by twofold when overexpressing eGFP-Trf1 WT (Fig 5H) but not when overexpressing the mutant versions of GFP-TRF1 (both single and triple TRF1 mutants in ERK1/2 phosphosites T328, T330, and T335; Fig 5H), which showed significantly higher amounts of TIFs which were comparable to that observed in control lines treated with ERKi (Fig 5H). These results indicate that the telomere defects induced by ERKi are mediated by TRF1 phosphorylation.

Next, to provide direct evidence for a role of ERK1/2 in TRF1 regulation, we genetically depleted ERK1/2 kinase in MEFs and analyzed TRF1 protein levels both by Western blot and nuclear

**Figure 4. ERK2, bRaf, and mTOR kinases phosphorylate TRF1 *in vitro*.**

A–D 1 or 2 μM of GST or GST-TRF1 was incubated with the indicated concentrations of mouse ERK2 kinase (A), human BRaf kinase (WT or V600E) (B, C), or mouse MEK1 kinase (D) in the presence of 5 μCi [γ-$^{32}$P]ATP. The mixture was resolved by SDS–PAGE followed by autoradiography.

E 1 μM of GST-TRF1 and 0.2 μM of mouse ERK2 kinase were incubated in the presence of ERK and MEK inhibitors.

F 2 μM of GST-TRF1 and 0.1 μM of human BRaf kinase were incubated in the presence of the bRaf inhibitors dabrafenib and vemurafenib.

G, H 1 or 2 μM of GST or GST-TRF1 was incubated with the indicated concentrations of human mTOR kinase (G) in the presence of the mTOR inhibitors rapamycin and Ku0063794 (H).

I–K Phosphopeptide peak intensity normalized to total TRF1 signal in samples containing only TRF1 or TRF1 plus ERK2 (I), TRF1 plus bRAF$^{WT}$ or bRAF$^{V600E}$ (J), and TRF1 plus mTOR (K); data are representative of *n* = 2 independent experiments.

L Schematic representation of TRF1 protein with the phosphorylation sites by ERK2, bRAF, mTOR, and AKT.

M–O Representative image (down) and quantification (up) of *in vitro* phosphorylation assays with the indicated GST-TRF1 wild-type or mutated forms in the presence of mouse ERK2 kinase. Data are representative of *n* = 4 independent experiments.

Data information: Data are represented as mean ± SEM. Significant differences using unpaired *t*-test are indicated by *$P < 0.05$, **$P < 0.01$, ***$P < 0.001$, ****$P < 0.0001$.

Source data are available online for this figure.

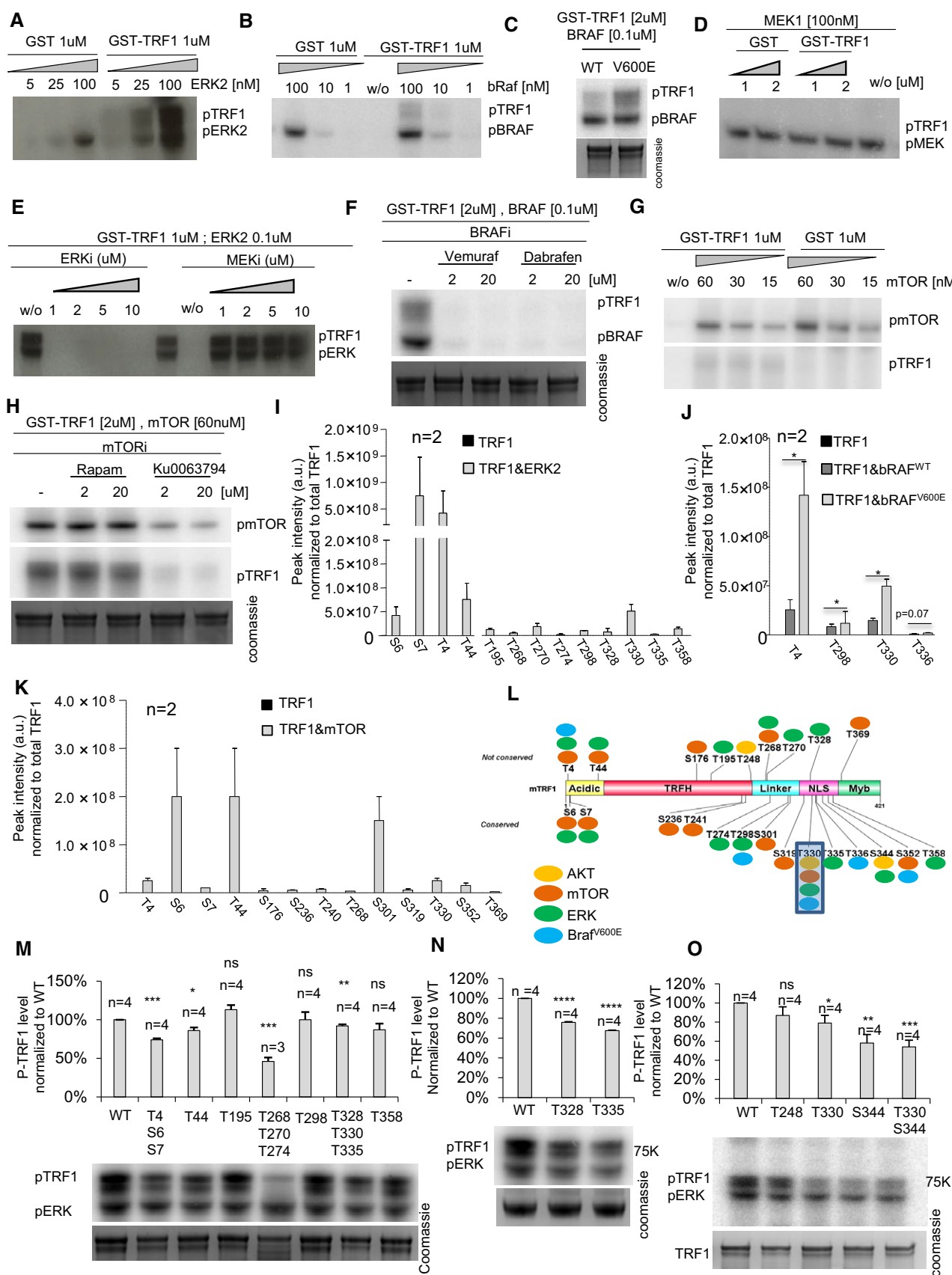

**Figure 4.**

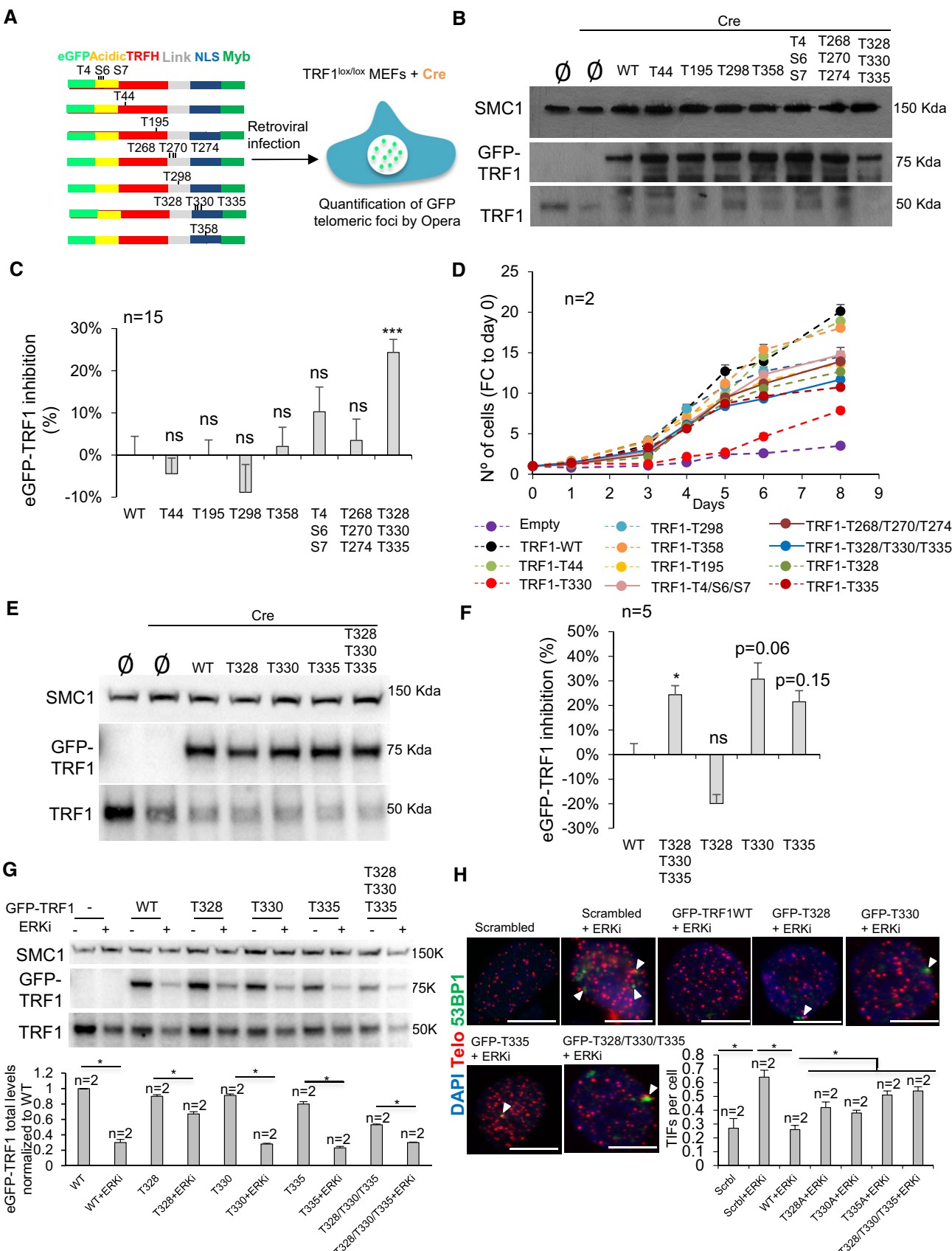

**Figure 5.**

◄

**Figure 5.  Phosphorylation of TRF1 residue T330 stabilizes TRF1 telomeric foci *in vivo*.**

A *Trf1*$^{lox/lox}$ *p53*$^{-/-}$ MEFs were transduced with eGFP-tagged *Trf1* WT or mutant alleles to overexpressed TRF1 depicted variants. Endogenous TRF1 was deleted by transduction with Cre recombinase. Opera High Content Screening (HCS) system was used to quantify the GFP spot intensity per cell.

B Western blot images of *Trf1*$^{lox/lox}$ MEFs with or without overexpression of eGFP-*Trf1* WT or mutant alleles followed by Cre recombinase transduction.

C Quantification of eGFP-TRF1 inhibition in *Trf1*$^{\Delta/\Delta}$ MEFs transduced with eGFP-*Trf1* WT or mutant alleles as indicated. Data are representative of *n* = 15 biological replicates

D Growth curves of *Trf1*$^{\Delta/\Delta}$ MEFs transduced with eGFP-*Trf1* WT or mutant alleles as indicated. Data are representative of *n* = 5 biological replicates

E Western blot images of *Trf1*$^{lox/lox}$ MEFs with or without overexpression of eGFP-*Trf1* WT or mutant alleles followed by Cre recombinase transduction.

F Quantification of eGFP-TRF1 inhibition in *Trf1*$^{\Delta/\Delta}$ MEFs transduced with eGFP-*Trf1* WT or mutant alleles as indicated. Data are representative of *n* = 5 biological replicates

G Western blot images of *p53*$^{-/-}$ MEFs with or without overexpression of eGFP-*Trf1* WT or mutant alleles followed treatment with ERKi. Data are representative of *n* = 2 biological replicates

H Representative images (above) and percentage (bottom) of telomeric and 53BP1 colocalizing foci (TIFs) per cells of *p53*$^{-/-}$ MEFs with or without overexpression of eGFP-*Trf1* WT and the indicated mutants upon treatment with the ERKi. White arrowheads: colocalization of telomeric and 53BP1. Scale bars, 10 μm. Data are representative of *n* = 2 independent experiments.

Data information: Data are represented as mean ± SEM. Significant differences using unpaired *t*-test are indicated by *$P < 0.05$, ***$P < 0.001$.
Source data are available online for this figure.

---

TRF1 fluorescence. We found that stable RNAi-mediated downregulation of ERK1/2 kinases resulted in a significant decrease in TRF1 protein levels (Fig 6A) as well as in TRF1 foci fluorescence compared to controls with a scrambled oligo (Fig 6B), thus demonstrating that genetic ERK depletion recapitulates the effects of chemical ERK inhibitors on TRF1.

Finally, we overexpressed eGFP-tagged versions of TRF1 WT, as well as single and triple TRF1 mutants in T328, T330, and T335 residues (Fig 6C and D). As expected, we found that stable RNAi-mediated downregulation of ERK1/2 kinases resulted in a significant decrease in TIFs (Fig 6E) compared to controls with a scrambled oligo (Fig 6B). Interestingly, the induction of TIFs as the result of ERK1/2 genetic downregulation was reduced to control levels when overexpressing eGFP-Trf1 WT (Fig 6E) but not when overexpressing the mutant versions of GFP-TRF1 (both single and triple TRF1 mutants in ERK1/2 phosphosites T328, T330, and T335) (Fig 6E), which showed significantly higher amounts of TIFs which were comparable to that observed in control lines treated with ERKi (Fig 6E). These results indicate that the telomere defects induced by ERK1/2 genetic downregulation are mediated by TRF1 phosphorylation.

## Combinatory treatments to avoid resistance to TRF1 inhibition in cancer

Finding novel TRF1 inhibitors not only is interesting to understand TRF1 biology but also to design rational combinatory treatments to

more effectively inhibit TRF1 levels in cancer and avoid appearance of resistance mechanisms.

In this regard, it is known that the bad prognosis of glioblastoma is mainly due to the existence of a group of cells with stem-like properties, also known as glioma stem-like cells (GSCs; Singh *et al*, 2004). These cells develop resistance to the treatments and are able to recapitulate the whole tumor, causing a strong recurrence (Bao *et al*, 2006). To model this phenomenon in mice, we injected patient-derived h676 and h543 GSCs into nude mice and treated them orally with our ETP-47037 PI3K inhibitor, previously shown by us to be a potent inhibitor of TRF1 foci ETP-47037 (Bejarano *et al*, 2017). One week after GSC injection, mice received oral administration of the vehicle as placebo or of the ETP-47037 inhibitor 5 days/week, every week until human end-point, and tumors were continuously followed up by caliper measurements. Treatment with the PI3K inhibitor ETP-47037 significantly slowed tumor growth in both h543 and h676 xenografts (Fig EV3A and B), in agreement with our previous findings (Bejarano *et al*, 2017). However, tumors became resistant approximately 1 month after treatment with the ETP-47037 inhibitor and grew until they reached the human end-point (Fig EV3A and B). To check whether the PI3K pathway was inhibited in these resistant tumors, we checked p-AKT and p-S6 levels by Western blot in h543 xenografts but we did not observe any changes between ETP-47037 and control treated mice (Fig EV3C), indicating that these tumors are able to reactivate PI3K pathway. In agreement

---

**Figure 6.  Genetic model of the telomeric role of ERK1/2-mediated TRF1 phosphorylation.**

A Western blot image (above) and quantification (bottom) of TRF1 protein levels upon genetic depletion of ERK1/2 in *p53*$^{-/-}$ MEF line. Data are representative of *n* = 3 independent experiments.

B Representative images (above) and quantification (bottom) of TRF1 telomeric foci in ERK1/2 RNA interfered *p53*$^{-/-}$ MEFs. Scale bars, 5 μm. Data are representative of *n* = 2 independent experiments.

C *p53*$^{-/-}$ MEFs were sequentially transduced with lentiparticles encoding short hairpins against ERK1/2 and retroparticles for eGFP-tagged *Trf1* WT or mutant alleles to overexpress TRF1-depicted variants in the absence of ERK1/2.

D Western blot image of eGFP-tagged and endogenous TRF1 protein levels upon genetic depletion of ERK1/2 in *p53*$^{-/-}$ MEF line.

E Representative images (above) and percentage (bottom) of telomeric and 53BP1 colocalizing foci (TIFs) per cell in *p53*$^{-/-}$ MEF with or without overexpression of eGFP-*Trf1* WT and indicated mutants upon genetic depletion of ERK1/2. White arrowheads: colocalization of telomeric and 53BP1. Scale bars, 10 μm. Data are representative of *n* = 2 biological replicates.

Data information: Data are represented as mean ± SEM. Significant differences using unpaired *t*-test are indicated by *$P < 0.05$, **$P < 0.01$.
Source data are available online for this figure.

►

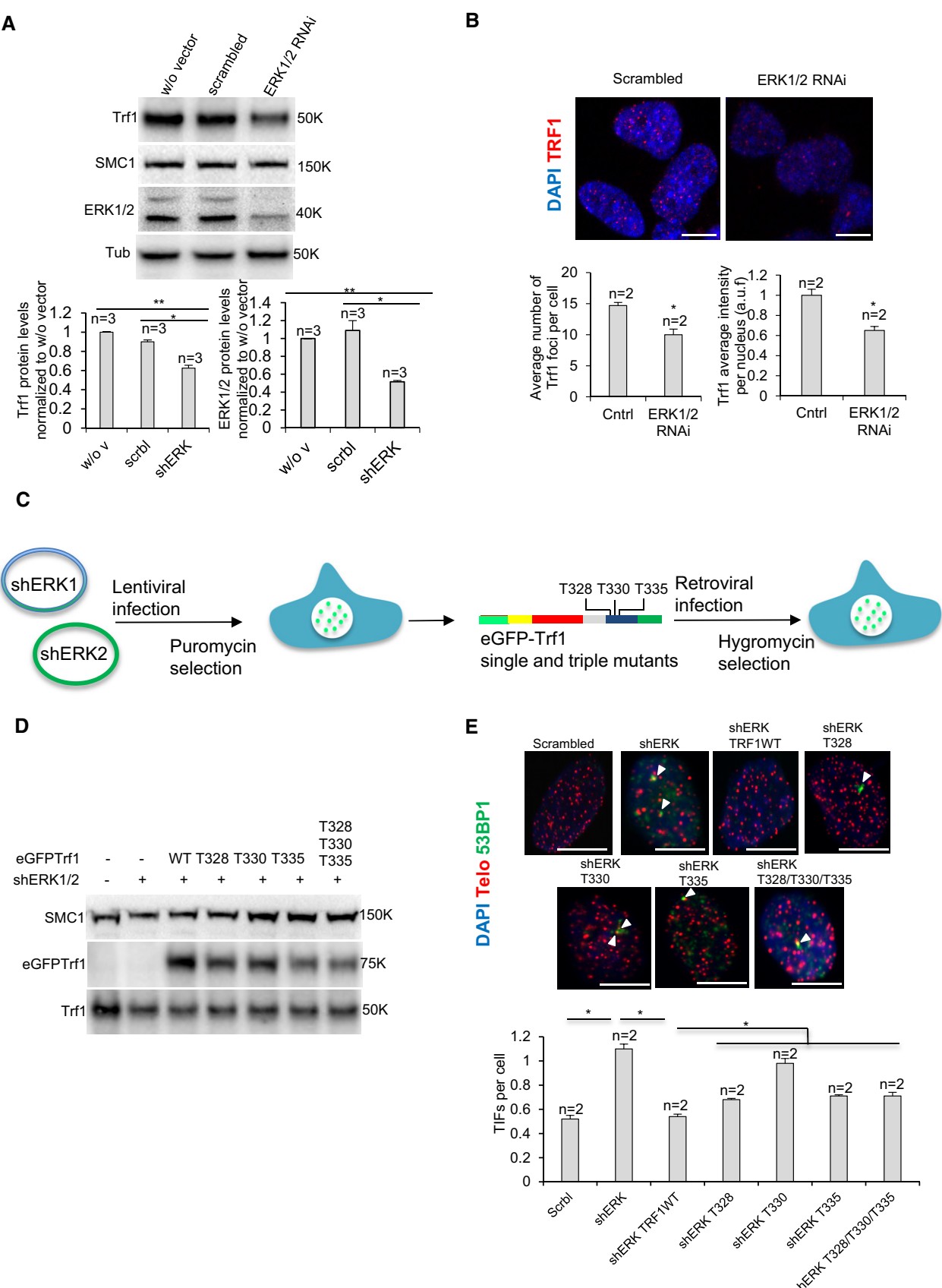

**Figure 6.**

with this, TRF1 immunofluorescence and Western blot analysis revealed that resistant tumors also had similar TRF1 levels than control tumors (Fig EV3D and E), suggesting that TRF1 re-expression was associated with tumor re-growth.

Thus, to more effectively block TRF1 expression and avoid resistance to individual drug treatments, we performed drug combination studies in GSCs in order to design new combinatory treatments based on TRF1 inhibition. In particular, we tested whether the PI3K inhibitors, already known to modulate TRF1, could show synergic effects with the compounds identified here, namely MEKi, ERKi, RTKi, CDKi, HSP90i, docetaxel, gemcitabine, and mTORi. To assess possible synergic effects between the different compounds, we used the GI50 (Table 1) as a reference point and designed a combinatorial matrix using different concentrations of the compounds (e.g., $2 \times EC_{50}$, $EC_{50}$, and $\frac{1}{2} EC_{50}$), studying all the possible combinations. Combination index is calculated to establish whether the combination is synergistic, additive or antagonistic, using the Chou–Talalay method (Chou, 2010). The lower combinatorial index indicates a better synergic effect between the compounds. Based on this combinatorial index, we selected the best combinations for further validations (Table 2). In particular, we selected the combination of PI3K inhibitors MEKi, ERKi, RTKi, HSP90i, docetaxel, and gemcitabine (Table 2). On the other hand, the combination of PI3Ki with mTOR and CDKi was excluded, as we did not see synergistic effects of the combination (Fig EV4A and B).

We further validated these synergic effects by performing a sphere formation assay treating the cells for one week with the selected concentrations (Table 2). Both number of spheres and the diameter of the spheres revealed that PI3Ki had significant synergic effect with MEKi, ERKi, RTKi, HSP90i, docetaxel, and gemcitabine (Figs 7A–F and EV4C–H).

We next set to address whether those compounds which showed the best synergic effect in combination with the PI3K inhibitors were also able to synergistically reduce TRF1 protein levels. To this end,

we treated the patient-derived h676 GSCs with the single agents or the combinations and we quantified total TRF1 levels by Western blot. We observed a clear decrease in TRF1 protein levels when we combined PI3Ki with RTKi, MEKi, or gemcitabine (Fig 7G,I,K). We also observed a minor decrease upon combination of PI3Ki with ERKi or HSP09i (Fig 7H and J). However, we did not observe a significant reduction in TRF1 protein levels upon combination of PI3Ki and docetaxel (Fig 7L).

### Combination of PI3K inhibitors with MEK inhibitors, docetaxel, and gemcitabine synergistically reduces xenograft growth of patient-derived GSCs

We next set to address whether the drug combination shown to synergistically decrease TRF1 levels *in vitro* was also able to synergize *in vivo* in human patient-derived xenograph models.

To this end, we injected primary patient-derived GCSs (h676) cells subcutaneously into nude mice and treated them with orally or IP administered compounds, as single agents or in combination with the PI3K inhibitor ETP-47037. One week after h676 GSCs injection, mice received oral or IP administrations as follows: (i) the PI3K inhibitor was orally administrated 5 days/week at a concentration of 50 mg/kg, until the human end-point; (ii) the MEK inhibitor PD0325901 was orally administrated 5 days/week at a concentration of 3 mg/kg, until the human end-point; (iii) the ERK inhibitor was orally administrated 5 days/week at a concentration of 50 mg/kg, until the human end-point; (iv) the RTK inhibitor Dasatinib was orally administrated 3 days/week at a concentration of 30 mg/kg, until the human end-point; (v) gemcitabine was intraperitoneally administrated at a concentration of 125 mg/kg, once a week, until the human end-point; (vi) docetaxel was intraperitoneally administrated at a concentration of 7.5 mg/kg, twice a week, until the human end-point; (vii) the HSP90 inhibitor was orally administrated 3 days/week at a concentration of 60 mg/kg, until the human end-point (see Materials and Methods). Note that all the listed compounds were administrated alone, or in combination with the PI3K inhibitor ETP-47037 and tumor growth was always compared to the placebo group.

Among all the possible combinations, the combination of the PI3K inhibitor ETP-47037 with ERKi, RTKi, and HSP90i resulted toxic in early stages of the treatment (Appendix Fig S5A), although we observed a synergism with ERKi (Fig 8A). On the other hand, the rest of the combinations—PI3Ki with MEKi, gemcitabine, and docetaxel—were well tolerated by the mice. In addition, tumor follow-up revealed that the PI3K inhibitor ETP-47037 also showed significant synergic effect with MEKi, gemcitabine, and docetaxel (Fig 8B–D), but not with RTKi or HSP90i (Appendix Fig S5B and C).

Table 2. Concentrations for synergic combinations in h676 GSCs.

| Combination | Concentration PI3Ki (nM) | Concentration Compound 2 (nM) |
|---|---|---|
| PI3Ki + HSP90i (Geldanamycin) | 0.2–0.4 | 0.005 |
| PI3Ki + docetaxel | 0.2–0.4 | 0.001 |
| PI3Ki + gemcitabine | 0.2–0.4 | 0.006 |
| PI3Ki + ERKi | 0.2–0.4 | 0.3 |
| PI3Ki + MEKi | 0.2–0.4 | 0.24 |
| PI3Ki + RTKi (Dasatinib) | 0.2–0.4 | 10 |

**Figure 7. *In vitro* combinatorial studies of PI3Ki with novel TRF1 inhibitory compounds.**

A–F   Representative images (left) and quantification (right) of number of spheres formed by patient-derived h676 GSCs 7 days after treatment with the indicated compounds as single agents or in combination. Scale bars, 100 μm. Data are representative of $n = 6$ biological replicates.

G–L   Western blot images (left) and TRF1 protein levels (right) measured in patient-derived h676 GSCs 24 h after treatment with the indicated compounds as single agents or in combination. Data are representative of $n = 3$ (combination in K), $n = 4$ (combination in G, L), $n = 5$ (RTKi in G, combination in H–J), docetaxel in L), $n = 6$ (DMSO in G–L, and ERKi in H), $n = 11$ (PI3Ki in G–J, L) biological replicates.

Data information: Data are represented as mean ± SEM. Significant differences using unpaired *t*-test are indicated by $*P < 0.05$, $**P < 0.01$, $***P < 0.001$.
Source data are available online for this figure.

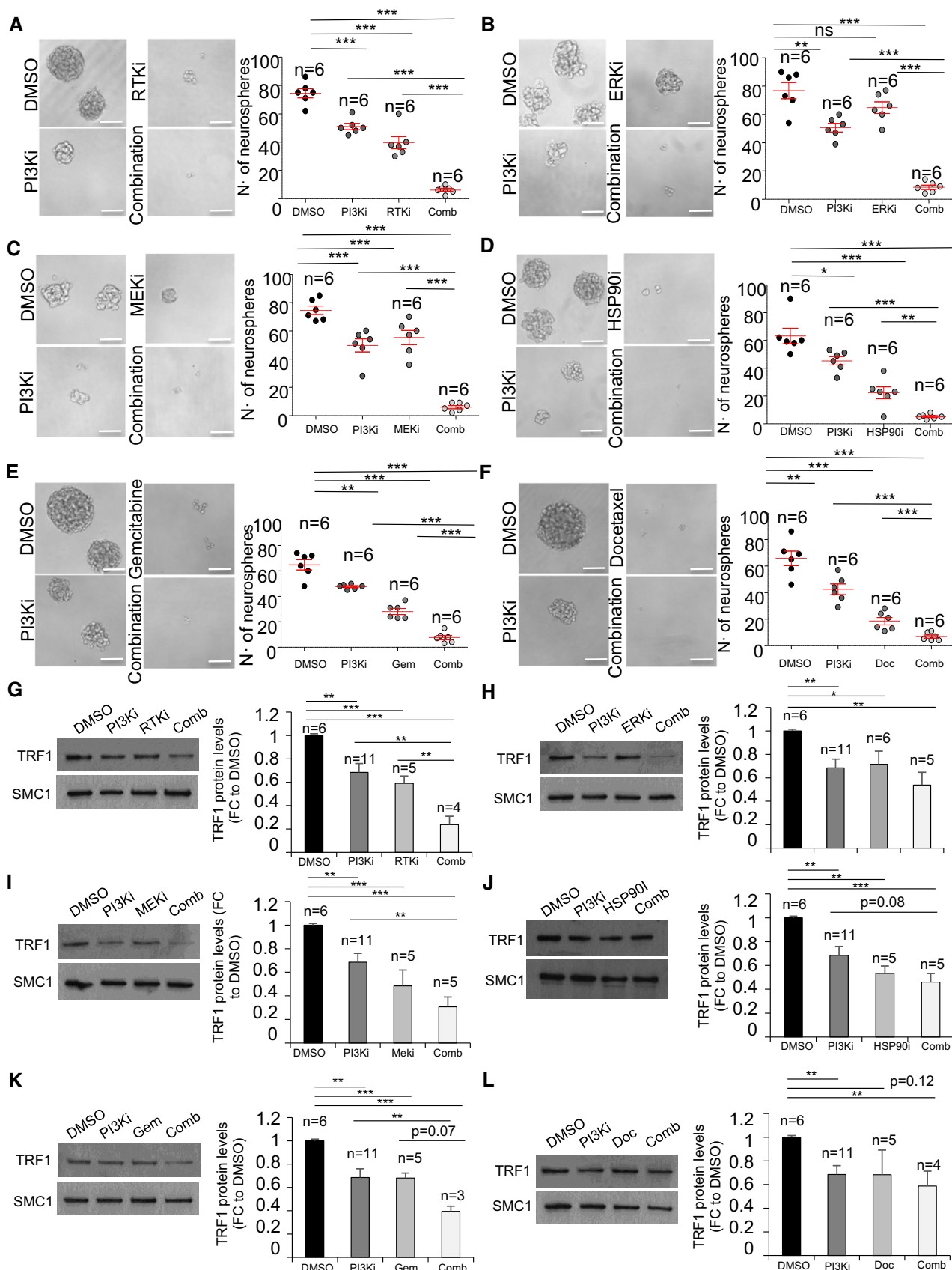

Figure 7.

We next set to address TRF1 levels in those combinations which showed a positive synergic effect, i.e., PI3Ki with ERKi, MEKi, gemcitabine, and docetaxel. We performed TRF1 immunofluorescence analysis in tumor samples treated with vehicle, single agents, or double agents. As shown here, long-term PI3Ki-treated tumors did not present a reduction in TRF1 levels (Fig 8E–H). In contrast, we observed a clear decrease in TRF1 nuclear fluorescence when the tumors were treated with combinations of PI3Ki and either ERKi or MEKi (Fig 8E–F), again demonstrating that these pathways modulate TRF1 levels independently. In the case of PI3Ki combination with docetaxel, we also observed a significantly lower TRF1 levels with the dual treatment (Fig 8G). However, docetaxel alone was also able to induce a strong decrease in TRF1 levels compared to vehicle-treated tumors (Fig 8G). Finally, the gemcitabine was not able to decrease TRF1 level neither alone nor in combination with PI3Ki (Fig 8H).

## Discussion

Telomere maintenance above a minimum length is essential to achieve an unlimited replicative potential, which makes telomere maintenance indispensable for the growth of cancer cells. In fact, more than 90% of human tumors aberrantly overexpress telomerase (Kim *et al*, 1994; Shay & Bacchetti, 1997; Joseph *et al*, 2010), while the remaining telomerase-negative tumors activate ALT (Bryan *et al*, 1997; Barthel *et al*, 2017). Thus, multiple studies have focused in telomeres as potential anti-cancer targets. In this regard, the most advanced compound is the telomerase inhibitor GRN163L, also known as Imetelstat (Harley, 2008; Joseph *et al*, 2010). However, telomerase inhibition in mouse models and in human clinical trials has shown some limitations, most likely due to the fact that a telomerase abrogation would only affect critically short telomeres, and tumors are heterogeneous in terms of telomere length (Chin *et al*, 1999; Greenberg *et al*, 1999; Gonzalez-Suarez *et al*, 2000; Parkhurst *et al*, 2004; Perera *et al*, 2008; Middleton *et al*, 2014).

Interestingly, not only telomerase, but also shelterin components are frequently altered in cancer. Our group and others have recently described that the TRF1-interacting protein POT1 is often mutated in familiar glioblastoma cases as well as other tumor types (Newey *et al*, 2012; Ramsay *et al*, 2013; Robles-Espinoza *et al*, 2014; Shi *et al*, 2014; Zhang *et al*, 2014; Bainbridge *et al*, 2015; Calvete *et al*, 2015). Also, we had recently reported that TRF1 is significantly upregulated in both mouse and human glioblastoma tumor samples (Bejarano *et al*, 2017). More importantly, we showed that TRF1 genetic deletion significantly impairs tumor growth in both

*p53*-deficient *K-RasG12V*-induced lung tumors and glioblastoma mouse models, in a manner that is independent of telomere length (Garcia-Beccaria *et al*, 2015; Bejarano *et al*, 2017). These facts highlight the possibility of alternative therapies to telomerase inhibition to impair telomere capping in cancer, by directly inhibiting the TRF1 shelterin component.

However, the mechanisms of TRF1 regulation in cancer are still poorly understood. In addition, novel compounds with the ability to downregulate TRF1 levels need to be developed. On this matter, our group performed an initial screening with the aim of identifying molecules and novel signaling pathways that modulate TRF1 binding to telomeres (Garcia-Beccaria *et al*, 2015). We identified several compounds which belonged to the PI3K family (Mendez-Pertuz *et al*, 2017). In particular, we found that both PI3K and AKT inhibitors significantly reduced TRF1 telomeric foci and this caused increased telomeric DNA damage and fragility (Mendez-Pertuz *et al*, 2017). Additionally, we found that TRF1 is phosphorylated at residues T248, T330, and S344 by AKT and that these modifications regulate TRF1 foci formation *in vivo* (Mendez-Pertuz *et al*, 2017). These findings uncovered an important functional connection between the Pi3K pathway and TRF1 regulation, thus connecting two of the major pathways in cancer and aging, namely telomeres and the Pi3K pathway (Mendez-Pertuz *et al*, 2017).

In order to identify additional signaling pathways that modulate TRF1 binding to telomeres, here we screened a CNIO collection of 114 anti-tumoral drugs, which are either FDA-approved or in clinical trials and which cover 20 of the 26 pathways included in Reactome database (see Materials and Methods). We identify novel drugs that can inhibit TRF1 in both lung cancer and glioblastoma cells, including inhibitors of some of the most deregulated pathways in cancer, namely, the Ras pathway (ERKi and MEKi), the cell cycle-related CDK inhibitor, the inhibitor of the chaperone HSP90, two chemotherapeutic agents (gemcitabine and docetaxel), and RTK and mTOR inhibitors. We show here that these drugs recapitulate the effects of *Trf1* genetic deletion, including DNA damage induction, telomere fragility, and reduction of stemness. Importantly, some of these drugs act independently of the Pi3K pathway, i.e., the Ras pathway components ERKi and MEKi, CDKi, gemcitabine, and docetaxel, underlying the importance of telomere capping for cancer cells.

We further demonstrate a role of the Ras pathway in regulating telomere protection. In particular, we show that both ERK2 and bRaf are able to phosphorylate TRF1 *in vitro*. Furthermore, we identify here 13 possible sites for TRF1 phosphorylation by ERK2, as well as four different putative sites for TRF1 phosphorylation by bRaf. As the ERK2 kinase is a downstream component of the pathway, we

---

**Figure 8. *In vivo* combinatorial studies of PI3Ki with novel TRF1 inhibitory compounds in patient-derived GBM xenograft models.**

A–D  Longitudinal tumor growth follow-up in mice injected with patient-derived h676 GSCs and treated with the indicated compounds in single agents or combination. n represents number of tumors, in (A): vehicle *n* = 16, PI3Ki *n* = 8, ERKi *n* = 16, combination *n* = 8; in (B): vehicle *n* = 16, PI3Ki *n* = 16, MEKi *n* = 8, combination *n* = 8; in (C): vehicle *n* = 8, PI3Ki *n* = 16, docetaxel *n* = 16, combination *n* = 8; in (D): vehicle *n* = 16, PI3K *n* = 16, gemcitabine *n* = 4, combination *n* = 4. P-values represent the mean of all the time points.

E–H  Representative images (top) and quantification (bottom) of TRF1 nuclear fluorescence in tumors treated with the indicated compounds as single agents or in combination. Scale bars, 10 μm. Data are represented as mean ± SEM. n represents number of tumors: in (E): vehicle *n* = 15, PI3Ki *n* = 10, ERKi *n* = 8, combination *n* = 2; in (F): vehicle *n* = 15, PI3Ki *n* = 10, MEKi *n* = 8, combination *n* = 6; in (G): vehicle *n* = 15, PI3Ki *n* = 10, docetaxel *n* = 8, combination *n* = 4; in (H): vehicle *n* = 15, PI3K *n* = 10, gemcitabine *n* = 8, combination *n* = 6.

Data information: Significant differences using unpaired *t*-test with Welch's correction are indicated by *$P < 0.05$, **$P < 0.01$, ***$P < 0.001$.

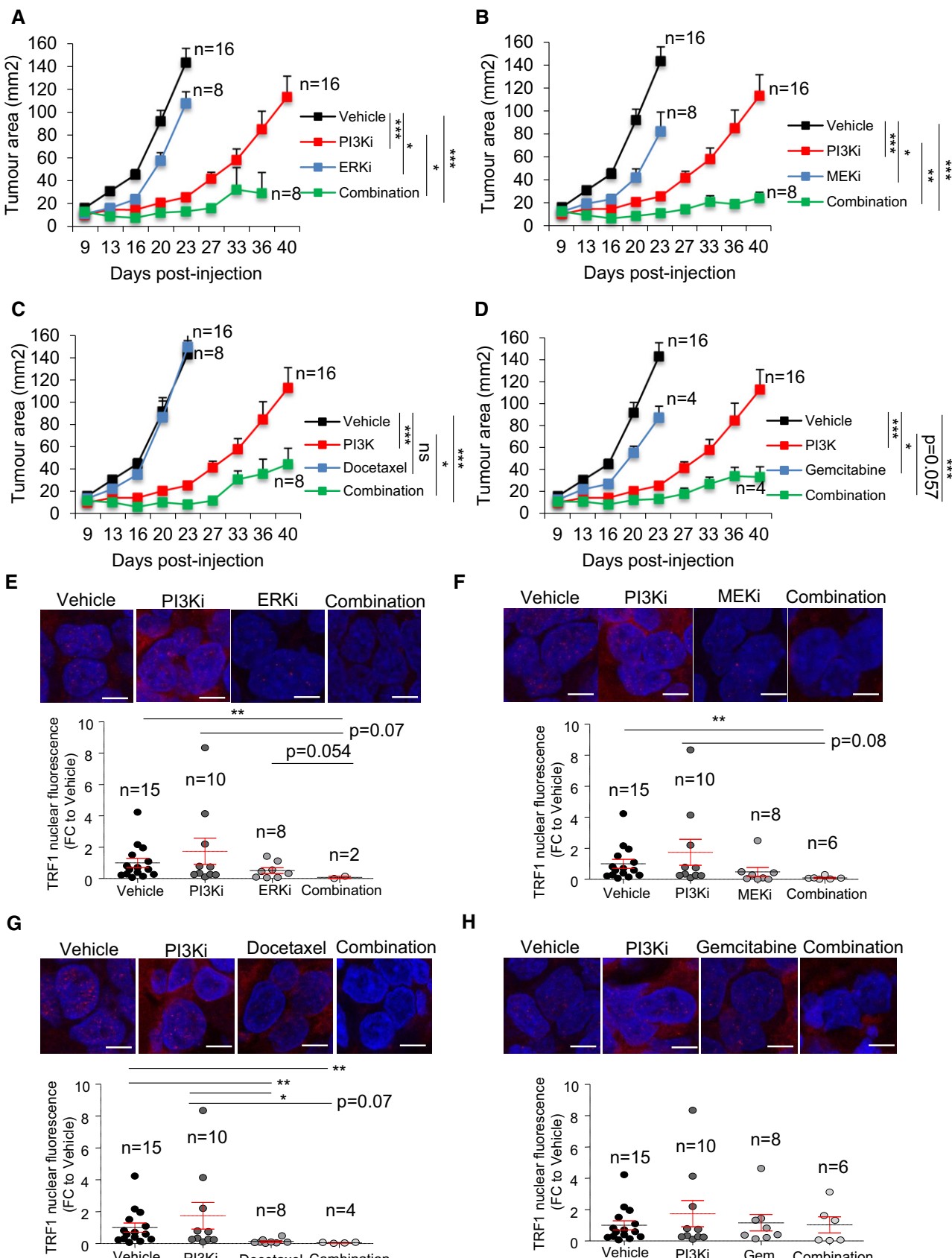

**Figure 8.**

further studied the *in vivo* role of the 13 possible phosphosites and showed that the mutation of T330 has a determinant role in TRF1 foci formation *in vivo*. Importantly, T330 is also phosphorylated by AKT (Mendez-Pertuz et al, 2017), bRaf, and mTOR kinases, suggesting that this site may be key residue for TRF1 modulation by different kinases. These unprecedented findings are in agreement with previous observations that ERK genetic deletion causes a reduction of TRF1 levels in embryonic stem cells (ES cells; Chen et al, 2015), as well as that TRF2, another shelterin protein, is targeted by ERK1/2 (Picco et al, 2016). Future studies warrant understanding regulation of other shelterin components throughout these and other post-transcriptional modifications. Interestingly, a previous study also showed that Ras overexpression can protect cancer cells from telomere dysfunction (Biroccio et al, 2013).

Finally, in the present work we use these new TRF1 regulatory pathways as a rational to discover novel drug combinations based on TRF1 inhibition, which effectively block resistance to individual drugs. In particular, it is known that the bad prognosis of many tumor types including glioblastoma is mainly due to the existence of a group of cells with stem-like properties, also known as glioma stem-like cells (GSCs; Singh et al, 2004). These cells develop resistance to the treatments and are able to recapitulate the whole tumor, causing a strong recurrence (Bao et al, 2006). By using patient-derived GSC xenograft models, here we show synergic effects of PI3K inhibitors, previously shown by us to inhibit TRF1 levels (Mendez-Pertuz et al, 2017), with ERKi, MEKi, gemcitabine, and docetaxel. In the case of gemcitabine, we observed differential effects *in vitro* and *in vivo*, which could be due to TRF1-independent effects of this drug.

In summary, here we describe that key cancer pathways, such as the Ras pathway, are important regulators of telomere capping through post-transcriptional modification of the shelterin component TRF1, essential for telomere protection. Furthermore, we demonstrate that these findings provide a new rational for the design of new combinatorial therapies based on TRF1 inhibition in cancer.

# Materials and Methods

### Mice

For xenograft experiments, 8-week athymic nude females were obtained from Harlan (Foxn1$^{nu/nu}$). Mice were maintained at the Spanish National Cancer Centre (CNIO) in accordance with the recommendations of the Federation of European Laboratory Animal Science Associations (FELASA) under specific pathogen-free conditions. All animal experiments were performed in accordance with the guidelines stated in the International Guiding Principles for Biomedical Research Involving Animals, developed by the Council for International Organizations of Medical Sciences (CIOMS), and were approved by the Ethical Committee (CEIyBA) from the CNIO. Along with those guidelines, mice were monitored in a daily or weekly basis and they were sacrificed in $CO_2$ chambers when the human end-point was considered.

### Cell culture and transfection

Mycoplasma-free patient-derived glioma stem cells (h543 and h676) were cultured in neurosphere medium from NeuroCult (Stem Cell Technologies Inc, Vancouver, Canada) supplemented with 20 ng/ml basic-FGF (RD Systems) and 10 ng/ml EGF (Gibco).

Mycoplasma-free $Trf1^{lox/lox}$ $p53^{-/-}$ MEFs and K-Ras$^{\Delta/LG12Vgeo}$ $p53^{-/-}$ tumor-derived CHA-9-3 cell line were cultured in DMEM supplemented with 10% fetal bovine serum (FBS). All cell lines and primary cultures were regularly tested mycoplasma-free.

For retroviral infection, supernatants were produced in 293T cells transfected with the ecotropic packaging plasmid pCL-Eco, the envelope plasmid pCMV-VSV-G and either pBabe-Cre and/or eGFP-TRF1 pWzl-Hygro (a gift from T. de Lange, Addgene plasmid #19834) using Fugene transfection reagent. Two days later, retroviral supernatants were collected and $Trf1^{lox/lox}$ p53$^{-/-}$ MEFs were infected with the corresponding retroviral supernatant at 12-h intervals.

For lentiviral infection, supernatants were produced in 293T cells transfected with the envelope plasmid pCMV-VSV-G, the ecotropic packaging plasmid psPAX2 and either pLKO.1 empty and/or pLKO.1 vectors expressing short hairpin RNA for either ERK1 or ERK2 (SIGMA) using Fugene transfection reagent. Two days later, lentiviral supernatants were collected and $p53^{-/-}$ MEFs were infected with the corresponding lentiviral supernatant at 12-h intervals.

### Screening of compounds

We tested 114 compounds belonging to a library of approved antitumoral drugs or in clinical phases to identify drugs that modulate TRF1 levels at telomere. The library covers 80% of the pathways described in Reactome. The number of inhibitors for each pathway was the following: cell cycle (2), cell–cell communication (10), cellular response to external stimuli (11), chemotherapeutics (2), DNA repair (2), extracellular matrix reorganization (3), gene expression (12), hemostasis (16), immune system (20), metabolism proteins (2), organelle biogenesis and maintenance (1), and signal transduction (26).

CHA-9-3 lung tumor cells were treated 24 h at 1 μM with the library of compounds, and TRF1 levels were assessed by immunofluorescence (See description below). For quantitative measurement of TRF1 foci levels, pictures of fixed cells were automatically acquired from each well by the Opera High Content Screening (HCS) system (Perkin Elmer).

The novel TRF1 modulators identified in the screening of compounds approved by the FDA or in clinical trials are the following:

ETP-50853—HSP90i (Geldanamycin)
ETP-45335—Docetaxel
ETP-45337—Gemcitabine
ETP-51634—Aurorai (Alisertib)
ETP-51801—RTKi (Dasatinib)
ETP-51799—PLK1i (GSK461364)
ETP-50537—mTor1/2i (KU-0063794)
ETP-50728—ERKi (SCH772984)
ETP-51667—MEKi (Selumetinib)
ETP-47306—CDKi (Flavopiridol)

We also tested the following compounds:

ERKi: GDC-0994, VRT752271
MEKi: Trametinib Cobimetinib PD-0325901 TAK-733
HSP90i: Debio_0932, 17-AAG, NVP-AUY922, NVP-HSP990
Taxol derivatives: ABT_751(E706), paclitaxel

## Phosphorylation assays

The Gateway™ technology (Thermo Fisher Scientific) was used to clone the full-length mouse TRF1 into the expression vector pDEST565, which adds two tags (6xHis and GST) at the N terminus of the encoded protein. Protein was expressed in Escherichia coli strain BL21(DE3), followed by purification with affinity chromatography using a $Ni^{2+}$ column (HisTrap FF crude, 17-5286-01, GE Healthcare) in an AKTA Prime system (GE Healthcare), and dialysis against 20 mM phosphate buffer pH 7.5, 200 mM NaCl, 2 mM TCEP.

One or two μM of GST or GST-TRF1 was incubated with human mTOR kinase (Millipore), human bRAF WT or V600E kinases (Millipore), or mouse ERK2 kinase (Millipore) in the presence of 5 μCi $[\gamma\text{-}^{32}P]$ATP. Note: For the analysis of BRAF and mTOR-dependent TRF1 phosphorylation, we used catalytically active "human" purified proteins consisting only in the catalytic C-terminal portion, since the murine BRAF and mTOR were not available. Nevertheless, albeit mouse and human mTOR proteins display 98.9% identity, the identity between their C-terminal kinase domains is 100%; thus, they exactly represent the same protein.

The following inhibitors were used: ERKi (SCH772984), MEKi (Selumetinib), BRAFi (vemurafenib and dabrafenib), mTORi (rapamycin and Ku0063794). The reactions were performed at 30°C for 1 and stopped by addition of Lamely buffer (6×). Samples were resolved in 4–12% SDS–PAGE followed by autoradiography.

## Neurosphere formation assays

Patient-derived h676 spheres were dissociated into single cells and seeded at a density of 100–200 cells/well in a 96-well plate. Neurosphere number was assessed after 7 days. Nikon Eclipse Ti-U microscope was used to take the pictures, and neurosphere diameter was measured using NIS Elements BR software.

## Generation of GST-TRF1 and eGFP-TRF1 mutant alleles

For site-directed mutagenesis, QuickChange XL II site-directed mutagenesis (Agilent Technologies) was used. In brief, PCRs were performed following the manufacturer's protocol with either the pDEST565-mTRF1 expression vector or the eGFP-TRF1 pWzl-Hygro retroviral vector as templates and the following purified mutagenic primers (Invitrogen):
mTRF1-T330A: 5′-GAACGAAGCAAGAACAGGAGCTCTTCAGTGTGAAACAAC-3′.
mTRF1-S344A: 5′-GGAAAGGAACCGAAGAACCGCTGGAAGGAATAGATTGTGT-3′.
mTRF1-T248A: 5′-CAACTTTTCTAATGAAGGCAGCAGCAAAAGTAGTGGAAAATGAGAAA-3′.
mTRF1-T4AS6AS7A 5′ATGGCGGAGGCGGTCGCAGCAGCGGCCCGG
mTRF1-T44A 5′CAGCTTCAGCTGGGGGCACCGAGAGAGATGGAG
mTRF1-T195A
5′GGTGATCCAGAATTTTACGCGCCTTTAGAAAGGAAG
mTRF1-T268AT270AT274A
5′CCAGATGCCGCCAACGCTGGAATGGACGCTGAAGTTG
mTRF1-T298A
5′GAAACTGAACCCTTAGTGGATGCAGTATCCTCAATAAG
mTRF1T328AT330AT335A

5′GAAGCAAGAGCAGGAGCTCTTCAGTGTGAAGCAACGATGG
mTRF1-T328A
5′GTAGGAACGAAGCAAGAGCAGGAACTCTTCAGTG
mTRF1-T335A
5′GAACTCTTCAGTGTGAAGCAACGATGGAAAGGAAC
mTRF1-T358A
5′GAGAATCAGCCAGACGCAGATGACAAAAGTGGACGC

PCR products were digested with Dpn I restriction enzyme to digest the parental (non-mutated) DNA for 1 h at 37 °C and then transformed into XL-10- Gold® ultracompetent cells. Individual colonies were grown and DNA extracted with QIAprep Spin Miniprep Kit (27106, QIAGEN). Mutations were confirmed by sequencing with a specific TRF1 primer 5′-TTCCACTCCCTTTTCCAACACT-3′. Finally, 50 ng of each mutant DNA was used to transform BL21 (DE3). Protein production and phosphorylation of the respective mutant GST-TRF1 protein were carried out following the same protocols described above.

## Identification of TRF1 phosphopeptides by LC/MS/MS analysis

1 μM or 2 μM of purified GST-TRF1 was incubated with 0.1 μM or 0.2 μM of either ERK2, mTOR, and BRAF in the proper kinase buffer (50 mM Hepes pH 7.9, 100 μM ATP, 10 mM MgCl 2, 5 mM DTT) in a total volume of 25 μl for 1 h at 30 °C. Protein samples were diluted with 9 M urea in 50 mM ammonium bicarbonate (ABC) and subsequently reduced and carbamidomethylated in 15 mM Tris(2-carboxyethyl)phosphine (TCEP) and 30 mM chloroacetamide for 45 min at 25°C protected from light. After diluting the urea to 2 M with 50 mM ABC, samples were digested overnight with Lys-C (1:50 enzyme/protein w/w), diluted to 1 M urea, and further digested with trypsin (1:50 enzyme/protein w/w) for 6 h at 37°C. Resulting peptides were desalted by homemade C18 Empore tips and analyzed by LC-MS/MS onto a LTQ-Orbitrap Velos instrument. The raw files were processed using the Proteome Discoverer 1.4.0.1 software. Fragmentation spectra were searched against the mouse UniProtKB/TrEMBL database (43,539 entries), supplemented with a homemade database comprising the contaminant proteins most commonly found in our assays, using Sequest as the search engine. The precursor and fragment mass tolerances were set to 20 p.p.m. and 0.5 Da, respectively, and up to two tryptic missed cleavages were allowed. Carbamidomethylation of cysteine was considered as fixed modification, while oxidation of methionine and phosphorylation of serine, threonine, and tyrosine were chosen as variable modification for database searching. The results were filtered to 1% false discovery rate (FDR) using percolator.

## Xenografts experiments

h676 and h543 patient-derived GSCs were dissociated using a 200-μl pipette and resuspended in NeuroCult medium and Matrigel in a 1:1 ratio in a concentration of 1,000 cells/μl. Foxn1[nu/nu] mice were subcutaneously injected with 100 μl of the cell preparation. The mice were randomized and treated with the compounds which were administrated as follows:
PI3Ki (ETP-47037): oral administration at a concentration of 75 mg/kg or 50 mg/kg 5 days per weeks until the end-point, starting 1 week after cell injection. Vehicle: 10% NMP / 90% PEG.

MEKi inhibitor (PD0325901): oral administration 5 days/week at a concentration of 3 mg/kg, until the human end-point. Vehicle: methylcellulose 0.5%: 0.2% Tween-80.

ERKi (VRT752271): oral administration 5 days/week at a concentration of 50 mg/kg, until the human end-point. Vehicle: 1% CMC.

RTKi (Dasatinib): oral administration 3 days/week at a concentration of 30 mg/kg, until the human end-point. Vehicle: 0.5% CMC 0.25% Tween-80.

Gemcitabine: intraperitoneal administration at a concentration of 125 mg/kg, once a week, until the human end-point. Vehicle: saline.

Docetaxel: intraperitoneal administration at a concentration of 7.5 mg/kg, twice a week, until the human end-point. Vehicle: saline.

HSP90i (Debio 0932): oral administration 3 days/week at a concentration of 60 mg/kg, until the human end-point. Vehicle: 30% Captisol.

Mice were weighed once a week for any sign in toxicity, and tumors were measured every 2–4 days in a non-blinded manner. Tumor area was determined by the following equation: $A = \pi * (a/2) * (b/2)$, where $a$ and $b$ are tumor length and width, respectively. The tumor growth curves were plotted until the last mouse died.

**Immunofluorescence analyses in cells and tissue sections**

For immunofluorescence analyses, we plated the cells in a proper density in cell culture µCLEAR plates (Greiner) and we fixed them in 4% formaldehyde in PBS. We incubated the cells with 0.25% Triton in PBS followed by 5% BSA in PBS.

Tissue sections were fixed in 10% buffered formalin (Sigma) and embedded in paraffin. After deparaffinization and citrate antigen retrieval, we incubated the slides with 0.5% Triton in PBS and blocked them with 1% BSA and 10% Australian FBS (Genycell) in PBS.

The antibodies were applied overnight in antibody diluents with background reducing agents (Invitrogen).

Primary antibodies: anti-Rap1 1:500 (BL735, Bethyl), anti-γH2AX Ser139 1:500 (05-636, Millipore), anti-TRF1 1:500 (BED5, Bio-Rad).

Images were obtained using a confocal ultraspectral microscope (Leica TCS-SP5). Quantifications were performed with Definiens software.

**Immunohistochemistry analyses in cells**

For immunohistochemistry analyses, cells were fixed in 10% buffered formalin (Sigma) and embedded in gelatine and paraffin. Immunohistochemistry was performed on deparaffinated cell sections treated with 10 mM sodium citrate (pH 6.5) cooked under pressure for 2 min. Slides were blocked with peroxidase, washed with TBS-Tween-20 0.5%, and blocked with fetal bovine serum followed by another wash.

Primary antibodies included the following: γH2AX Ser139 (Millipore), 1:150.

Slides were then incubated with secondary antibodies conjugated with peroxidase from DAKO.

Olympus AX70 microscope was used to take the pictures. The percentage of positive cells was counted by eye.

**The paper explained**

**Problem**

Eukaryotic chromosome ends, known as telomeres, are considered a universal anti-cancer target as telomere maintenance is essential to sustain cancer cell growth. Telomeres are protected by the so-called shelterin complex, and thus, targeting of the shelterin complex may be an effective anti-cancer strategy. We have previously demonstrated that genetic and chemical inhibition of the shelterin protein TRF1 impairs tumor growth in mouse models of lung cancer and glioblastoma, including patient-derived xenograft models. However, resistance to TRF1 chemical inhibitors eventually occurs, resulting in re-expression of the TRF1 protein and tumor growth. To find novel TRF1 inhibitors, which could be used in combination therapies to effectively block cancer growth, here we screened a collection of FDA-approved drugs and drugs in clinical trials.

**Results**

We found that inhibition of several kinases of the Ras pathway, including ERK and MEK, recapitulates the effects of Trf1 genetic deletion, including induction of DNA damage at telomeres and inhibition of cancer stemness. We further show that TRF1 is phosphorylated by ERK in thirteen different residues. We identify three TRF1 phospho-sites, whose ERK-dependent phosphorylation is required for proper telomere protection. Finally, we explore novel drug combinations based on TRF1 inhibition, with the aim of effectively blocking potential resistance to individual drugs in patient-derived glioblastoma xenograft models.

**Impact**

Glioblastoma is the most aggressive brain tumor, and despite treatment with chemotherapy, radiation, surgery-based combined treatments, it very frequently relapses. Thus, new therapeutic approaches are needed to effectively block glioblastoma growth. We have shown that TRF1 inhibition is a potent way to impair glioblastoma growth. Here, we show that multiple cancer pathways, including the Ras pathway, are potential therapeutic targets to inhibit TRF1 in cancer. Our findings further show the potential of combination therapies based on TRF1 inhibition as a promising therapeutic strategy to overcome drug resistance and effectively block glioblastoma growth.

**Quantitative telomere fluorescence *in situ* hybridization (Q-FISH) on metaphase spreads**

For metaphase preparation, cells were treated overnight with 0.1 µg/ml colcemid. Cells were incubated with hypotonic solution (0.4% KCl, 0.4% sodium citrate) and fixed with cold methanol/acetic acid (3:1). On the final steps, cells were spread on glass slides. For quantitative telomere fluorescence *in situ* hybridization (Q-FISH), slides were dehydrated with increasing concentrations of EtOH (70%, 90%, 100%) and then incubated with the telomeric probe for 3.5 min at 85°C followed by 2 h RT incubation in a wet chamber. After the incubation, the slides were washed with 50% formamide and 0.08% TBS-Tween.

Analysis of MTS signals was performed by superposing the FISH telomere image and the DAPI image.

**Western blots**

Nuclear Cytosolic Fractionation Kit (BioVision) was used to obtain protein extracts. Protein concentration was determined using the Bio-Rad DC Protein Assay (Bio-Rad). Up to twenty

micrograms of nuclear protein extracts was separated in SDS–polyacrylamide gels by electrophoresis. After protein transfer onto nitrocellulose membrane, the membranes were incubated with the indicated antibodies. Antibody binding was detected after incubation with a secondary antibody coupled to horseradish peroxidase using chemiluminescence with ECL detection KIT (GE Healthcare).

Primary antibodies: anti-TRF1 1:1,000 (BED5, Bio-Rad), anti-TRF1 1:500 (homemade), anti-SMC-1 1:2,000 (Bethyl), anti-AKT1 1:500 (Millipore), anti-p-AKT 1:500 (Ser473, Cell Signaling Cell Signaling Technology), anti-S6 1:500 (Cell Signaling Cell Signaling Technology), anti-p-S6 1:500 (Ser240/244, Cell Signaling Technology), anti-TIN2 1:1,000 (Abcam), anti-RAP1 1:1,000 (Bethyl).

For the quantification, protein-band intensities have been quantified by densitometric analysis with ImageJ software. The Trf1 total levels have been normalized versus SMC1 and the mean of the Trf1/SMC1 ratio deriving from at least 3 different replicates has been used to generate the chart.

### Combinatorial studies

For the combination studies, the concentration that produces 50% of the inhibition (EC50) in the growth of glioma stem cells is calculated. Using this concentration as a reference point, a combinatorial matrix is designed using different concentrations of the compounds (e.g., 2x EC50, EC50, and ½ EC50) to study all the possible combinations. Combination index is calculated to establish if the combination is synergistic, additive, or antagonistic based on the Chou–Talay method (Chou, 2010).

### Quantification and statistical analysis

Immunofluorescence quantifications were performed with Definiens software, and immunohistochemistry quantifications were performed by direct cell counting. Western blot protein-band intensities were measured with ImageJ software and normalized against the loading control.

Unpaired Student's *t*-test (two-tailed) or chi-square was used to determine statistical significance. *P*-values of less than 0.05 were considered significant. *$P < 0.05$, **$P < 0.01$, ***$P < 0.001$. Statistical analysis was performed using Microsoft® Excel 2011. All *in vitro* experiments were repeated at least twice (biological replicates) with two or more technical replicates. The exact *P*-values are listed in Appendix Table S2.

**Expanded View** for this article is available online.

## Acknowledgements

We thank the Confocal Microscopy, Protein Engineering, Mass Spectrometry, Comparative Pathology, and Mouse Facility Units at CNIO. MAB laboratory is funded by SAF2013-45111-R from MINECO, *Fundación Botín* and Banco Santander, Worldwide Cancer Research 16-1177. LB is a fellow of the La Caixa-Severo Ochoa International PhD Programme.

## Author contributions

MAB conceived the idea; MAB, LB, and GB designed the experiments. LB, GB, and JL performed the experiments. MAB, LB and GB wrote the manuscript. RS aided with mice treatments. EGC, SM, CBA, and JP performed the initial screening and provided the chemical inhibitors. JMT performed TRF1 protein purification.

## Conflict of interest

The authors declare that they have no conflict of interest.

## For more information

(i) https://www.cnio.es/en/research-innovation/scientific-programmes/molecular-oncology-programme/telomeres-and-telomerase-group/.

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
