## [Review Process File · EMBO Molecular Medicine]

Multiple cancer pathways regulate telomere protection

Leire Bejarano, Giuseppe Bosso, Jessica Louzame, Rosa Serrano, Elena Gómez-Casero, Jorge Martínez-Torrecuadrada, Sonia Martínez, Carmen Blanco-Aparicio, Joaquín Pastor and María A. Blasco

Review timeline:	Submission date:	8 January 2019
	Editorial Decision:	25 February 2019
	Revision received:	12 April 2019
	Editorial Decision:	3 May 2019
	Revision received:	10 May 2019
	Accepted:	14 May 2019

Editor: Lise Roth

Transaction Report:

1st Editorial Decision

25 February 2019

Thank you for the submission of your manuscript to EMBO Molecular Medicine. We have now heard back from the referees whom we asked to evaluate your manuscript.

As you will see from the reports below, the 3 referees mention the interest of the study, but they also have fundamental concerns that should be addressed in a major round of revision of the present manuscript, so that the data fully support the conclusions. In particular, the causality of TRF1 inhibition on the observed effects should be clearly established, and better quantification of the results throughout the manuscript should be performed.

Addressing the reviewers concerns in full will be necessary for further considering the manuscript in our journal. EMBO Molecular Medicine encourages a single round of revision only and therefore, acceptance or rejection of the manuscript will depend on the completeness of your responses included in the next, final version of the manuscript.

Please also contact us as soon as possible if similar work is published elsewhere. If other work is published, we may not be able to extend the revision period beyond three months.

I look forward to receiving your revised manuscript.

***** Reviewer's comments *****

Referee #1 (Remarks for Author):

This manuscript describes the identification and characterization of drugs reducing the level of TRF1. Several MAPK inhibitor were identified as TRF1 downregulating compounds leading to telomere dysfunction, impaired stemness and reduced tumorigenicity, when used as single agent and in combination. Accordingly, the authors identify the TRF1 sites phosphorylated by ERK1/2. The finding that TRF1 is regulated by the MAPK pathway is interesting for both our understanding of telomere regulation in general and the role of TRF1 in cancer therapies. However, the results appear preliminary regarding the importance of TRF1 in the anti-cancer effects of MAPK inhibitors. Indeed, these drugs have numerous cellular targets and the relative importance of TRF1 inhibition was only inferred from correlative observations. Thus, it is mandatory that the authors demonstrate that the effects they observe are really caused by TRF1 reduction and phosphorylation defects. To this end, they must test whether restoring a physiological level of WT TRF1 and phosphorylation mutants will rescue or not telomere dysfunction, DNA damage, stemness and tumorigenicity.

Additional comments:

Figure 2A Does the increased number of TIF stem from a specific telomere effect of the compounds or a general increase in DNA damage, including telomere among others. Thus the authors must normalize their TIF results to the total amount of DNA damage

Figure 2B; it is the reverse problem than in Figure 2A, an increase in DNA damage is not enough to conclude about any effect on TRF1 or telomere.

The authors should quote a previous publication relating that TRF2, another shelterin subunit, is targeted by ERK1/2 (Picco et al, Oncotarget, 2016).

Referee #2 (Remarks for Author):

In the current manuscript by Bejarano et al., the authors screened for TRF1 inhibitory drugs using a collection of FDA approved drugs and drugs and investigated the effect of these drugs on TRF1 expression. Their experiments revealed that the Ras pathway and PI3K pathway was associated with TRF1 expression by regulation of phosphorylation. In addition, the authors also found that these inhibitors suppressed tumor growth on patient-derived glioma stem cells, especially by a combination of PI3K inhibitors with MEK inhibitors and Gemcitabine. Considering these results, their findings suggest that TRF1 inhibition may be novel combinatorial therapies in cancer. Although their results are interesting and contribute to the understanding of the roles of TRF1 in cancer biology, some of the key conclusions are not fully supported by the present experiments, as outlined in more detail below. Therefore, I feel that this has to be revised before it can be considered for publication in EMBO Molecular Medicine.

Major concerns:

1. To test if TRF1 inhibitors (PI3Ki, mTORi, RTKi, ERKi, MEKi, Aurorai, CDKi, PLK1i, Docetaxel, Gemcitabine, and HSP90i) effect on TRF2, TPP1, POT1, TIN2 and RAP1, the authors should show the results of immunoblot analysis of these proteins in CHA-9.3 mouse lung cancer cells treated with TRF1 inhibitors.
2. Page 17, line 417-423: I feel the paragraph has a contradiction. The authors described that the combination of the PI3K inhibitor with Docetaxel resulted toxic in the early stages of the treatment in line 417-419. While the authors described that the combination of the PI3K inhibitor with Docetaxel was well tolerated in the mice in line 419-421. Considering Supplementary Figure5A, I think the combination of the PI3K inhibitor with Docetaxel has toxic in the mice.

Minor concerns:

1. Please check all the typo.
For example,
Page 13, line 330: "TRF1-335" should be corrected with "TRF1-T335"

Referee #3 (Remarks for Author):

General comments:

In this paper, by screening of 114 molecules, the authors identified new molecules to inhibit the telomeric localization of the TRF1 protein (or its expression) and analyzed the implication of such inhibitors (alone or in combination) on tumor growth. The data identified new molecular pathway implicated in TRF1 expression and TRF1 phosphorylation that could impact tumor growth and cancer stemness. More precisely, the author has performed a screen of 114 molecules to study the inhibition of TRF1 expression using Opera screening system. They apply the 114 molecules on CHA-9.3 mouse lung cancer cells line and identified 9 hits that has been secondly tested on same cells by WB. Identified hits are also tested on human Glioma Stem Cells (GSC) with similar effect for most the hits mainly affecting PI3K and Ras pathway and leading to a synergistic effect on tumor growth inhibition and stemness inhibition.

This paper can bring interesting novelty in the field and for a more general reader. The introduction clearly place the subject of the article into the general and specific context of the field. The importance of the main data obtained in this paper is also well placed in relation to the field of study and the potential medical consequences. However, there is some limitations in the interpretation and the analysis of the data leading to some overstatement or misleading conclusions (as detailed below). The experiment are technically well done but for some of the figures, the strength of the evidence is not in adequacy with the conclusions drawn as detailed below.

Major concerns:

1. The figure 1 present the results for 9 hits, confirmed on the 2 cell lines in WB and IF. Excepted the Sup Fig 1A showing the Reactome analysis, there is no information of the impact of all the other drugs tested (105 molecules?), even in sup fig or Table. At least a table with the list of the drugs and the quantification of the primary screen should be shown to know why and how the selected the 9 hits.
2. In fig 1C, the DMSO control seems to be loaded with more materials than the other lanes for the first and second panel. Moreover, the duplicate of the third panel is not consistent with less quantity of TRF1 in the right lane compared to the left. The quantity of TRF1 in the right control seems as low as Gem treated cells for instance. How the authors have quantified and normalized the data to compare all the lanes? It is not indicated in method section
3. Since the authors focused their work on glioblastoma for the rest of the paper, why they used the mouse lung cancer cell for the screening?
4. It appears that CHA93 treatment with the compounds increased the number of cells with TIFs (Fig 2A). It sounds risky to interpret the increase of TIF+ cells after mTOR inhibition "as a trend" as proposed by the authors (line 197-198). To the reviewer point of view, mTOR treatment do not change the proportion of TIF+ cells (15% in control to 18% in mTOR treated). It does not represent a trend to increase. In addition, for TIF counting, why the authors represent the number of cells with at least one TIF? Do they try to count the mean number of TIF per nucleus? In this case, do the mTOR treatment increase DNA damage at telomere? In Fig2B, the authors analyzed the total DNA Damage by quantifying the signal for gH2AX IHC and consider the increase of total DDR as a validation with GSC of the TIF results performed on mouse lung cancer cells. It seems difficult to directly compare these 2 experiments and to use this comparison as a validation. TIF in GSC seems to be more appropriate? Do they try to quantify the signal only for gH2AX in the Fig2A and not only the TIF? In this case, do they find a similar increase of gH2AX they saw in 2B (GSC) after mTOR inhibitor treatment with CHA93?
5. The response for RTKi seems to be very variable depending the assay. The dose response in 3A for h676 is very flat and RTKi do not seems to have a strong effect, that is consistent with Table 1 where the highest GI50 is obtain for RTKi. However, the effect of RTKi on TRF1 level is the strongest (Fig1C histograms on the right) or AKT phosphorylation (SF2) is the strongest and the effect on TIF (2A) and total gH2AX (2B) is intermediate. Can the authors comment on that?

6. Concerning mouse TRF1 phosphorylation analysis, why the authors performed the experiment using mouse TRF1 with mouse ERK2, mouse MEK1, human bRaf or human mTOR? Why they do not use mouse bRaf and mouse mTOR?

7. The authors claim in the text that the triple mutation T328/T330/T335 is associated with a significant reduction of TRF1 phosphorylation (line 299). Looking at the graph in the figure 4M, we can see a very low decrease but seems to be significant (two stars). Is this effect really significant? At least it seems to be significant but very moderated?

8. In fig5B, endogenous TRF1 is floxed and eGFP TRF1 WT or mutant is re expressed. The authors claimed that endogenous *trf1* is deleted (line 310) but the WB do not sustain this conclusion. The 2 controls are not homogenous: the second control is as low as the TRF1 deleted condition or lower (ie 2nd control and T298 or T4S6S7) and TRF1 WT expression is observed all the condition except for the T328T330T335 mutant. Why all the sample do not look like the T328 T330 T335 mutant? Is the TRF1 deletion complete? Is really a homozygote *lox/lox* *trf1* mutant?

The authors claims in Fig 5C that only the triple T328 T330 T335 mutant is able to inhibit TRF1 telomeric foci: "Quantification of nuclear eGFP spot fluorescence in *Trf1*^{-/-} MEFs showed that the triple mutant T328/T330/T335 was the only mutant which showed a significant decrease in the intensity of TRF1 telomeric foci compared to MEFs expressing wild-type TRF1 (Fig. 5C)". However, in line with the previous comments, the triple T328 T330 T335 is the only one showing a total deletion of endogenous TRF1. One interpretation could be that the remaining endogenous TRF1 interfere with the eGFP TRF1 to bring it at telomere. The conclusions of the authors seem overstated. In addition, only the quantification of the eGFP telomeric foci is shown and no images are presented, even in sup. Data should be shown.

9. The authors claimed that the triple T328 T330 T335 is the one showing the highest proliferation defect (Fig5E) and claim that it is in agreement with data from fig 5C showing the inhibition of telomeric foci. Again, the data do not sustain this conclusion since

- o the inhibition of proliferation of the triple T328 T330 T335 is very similar to the one of T268 T270 T274 or T4 S6 S7. Is there any statistical difference between these 3 mutants in terms of proliferation?
- o The T328 T330 T335 mutant is the only one where endogenous TRF1 seems to be fully deleted (see previous comment)
- o The effect of simple mutant T195 or T298 seems to be similar to the triple. Is the experiment performed in the same time? The black and violet curves is identical in 5D, 5E and 5G suggesting that all the mutant tested are performed in the same experiment. Can the data be plotted on the same graph and statistically analyzed to compare any differences?

10. Similarly to what they did in Fig 5B-E, the authors want to analyze the effect of single mutation T328, T330 and T335 (Fig5F and G). There is no information available concerning the single mutant T328, T330 and T335 for the expression of endogenous TRF1 like in 5D. Do the endogenous TRF1 is totally deleted? The full interpretation of the effect of single mutant compared to the triple mutant concerning the proliferation or telomeric foci cannot be performed without this WB. In addition, as for Fig5C, images related to Fig5G are missing.

11. Concerning data on tumor growth experiment and ETP47037 treatment (Fig S3), the authors said that the treatment start one week after GSC injection. DO the tumor are already palpable? DO they try to treat after the establishment of the tumor and not before? This could change the interpretation of the data.

12. The authors measured TRF1 expression level at the end of the tumor growth experiment and they do not see any difference between treated and not treated, likely due to the re expression of TRF1. But their conclusion is overstated. S3D and S3E do not "indicate" but "suggest" that the regrowth is associated to TRF1 re expression (line 362). Do they try to compare the TRF1 expression level in tumors treated and untreated at days 20 or day 30 when there is a strong difference of growth? In this case, do TRF1 is really less expressed? It could be also because the treatment select the cells with the highest expression of TRF1.

13. The data on neurospheres are convincing (6A-F and sup 4C-H). The quantification of neurospheres could be more appreciated if presented in box plot with all point as the authors did for

Sup Fig 4 for neurosphere diameter. Concerning WB for TRF1 after treatment, the quantification is wired on some aspect. For instance, the blot in 6H give the feeling that comb treatment is really synergistic and induce a strong decrease compared to control but the quantification does not sustain this decrease. Moreover, axis is not labelled. On the other hand, the quantification of the comb treatment in 6J is similar to the one in 6H but the blot signal seems to be higher. Are the authors be sure of the quantification? The decrease of TRF1, except for 6G and 6I is not synergistic while the effect on neurosphere seems to be synergistic. How the author can conciliate this discrepancy?

14. The effect of comb. Treatment on tumor growth is clear and appears synergistic (Fig7A-D and SF5). The author do not precise the number of mice per condition and should indicate it. It is not clear for the comb treatment that give toxicity how the author manage to perform the experiment: if 100% of the mice are dying for toxicity as shown in Sup F5A for PI3Ki + RTKi, how they succeed to obtain the data in 7A ?

15. In vitro and in vivo data are opposite in some cases like gemcitabine treatment. While gemcitabine alone or in comb with PI3K do not decrease TRF1 expression level in tumor sample, gemcitabine decrease TRF1 expression (Fig1), increase TIF and total gH2AX (fig 2), decrease number and diameter of neurospheres (fig3 and SF4), in synergy with PI3K inhibitor (fig 6E and 6K) and synergistically decrease tumor growth (fig 7C). How the author can explain this discrepancy?

Minor concerns:

16. It is not clear why the authors perform initially their screen on a mouse lung cancer cell line and then perform all the rest of the study on GSC using either H676 cells or H543 or both.

17. The authors claim that they demonstrate "an unprecedented role of the Ras pathway in telomere protection" (line 129-130 and 493). However, it has already been shown that Ras pathway could be implicated in telomere protection since it has been shown that Ras overexpression protect cancer cells from DDR induced by TRF2 dysfunction through a IL-6 dependent mechanism (Biroccio et al 2013 NCB). It could be interesting to confront these data and to include it in the discussion.

18. The legend of the fig1 is not complete and the annotation of the panel of the Fig 1 is misleading. What are the cells used for the quantification of TRF1 in the 2 histograms of the Fig1 just above the scheme of signaling pathway? Is it H676?

19. All over the text, authors refer to a "STAR method" like line 159 that is not formally present in the manuscript.

20. In Fig 3 and Table 1, the authors determined using the H676 and H543 GSC whether the treatment with the inhibitors alter the stemness of human GSC since they showed previously that TRF1 OE in GSC increase stemness. But, there is some unclear information: TRF1 inhibition decrease stemness in the 2 cell lines tested. why the authors do not perform the DDR analysis on H543 cells? Then, in the Table 1, there is probably a typo error. Is it really h676 or h767 cells used?

21. Various typo problems or edition problems (line 327; line 230; star methods everywhere...)

22. Line 427: do the authors really refers to fig S4D and E?

1st Revision - authors' response

12 April 2019

Detailed Answers to Referee #1:

This manuscript describes the identification and characterization of drugs reducing the level of TRF1. Several MAPK inhibitors were identified as TRF1 downregulating compounds leading to telomere dysfunction, impaired stemness and reduced tumorigenicity, when used as single agent and in combination. Accordingly, the authors identify the TRF1 sites phosphorylated by ERK1/2. The finding that TRF1 is regulated by the MAPK pathway is interesting for both our understanding of

telomere regulation in general and the role of TRF1 in cancer therapies. However, the results appear preliminary regarding the importance of TRF1 in the anti-cancer effects of MAPK inhibitors. Indeed, these drugs have numerous cellular targets and the relative importance of TRF1 inhibition was only inferred from correlative observations. Thus, it is mandatory that the authors demonstrate that the effects they observe are really caused by TRF1 reduction and phosphorylation defects. To this end, they must test whether restoring a physiological level of WT TRF1 and phosphorylation mutants will rescue or not telomere dysfunction, DNA damage, stemness and tumorigenicity.

ANSWER: We are very thankful to the reviewer #1 for considering that **“the finding that TRF1 is regulated by the MAPK pathway is interesting for both our understanding of telomere regulation in general and the role of TRF1 in cancer therapies”**.

We fully agree with the reviewer statement that **“it is mandatory that the authors demonstrate that the effects they observe are really caused by TRF1 reduction and phosphorylation defects” and we have now included extensive new experimentation to address this**. However, first we would like to point out the following experiments already included in the manuscript that indicate that the observed effects are caused by TRF1 reduction. First, we show that TRF1 chemical inhibition by using mTOR and MEK inhibitors specifically induces DNA damage at telomeres as indicated by an increase in the so-called telomere-induced foci or TIFs (see **Fig. 2A**) as well as by an increase in multiple telomeric signals (MTS; see **Fig. 2C**), both well established hallmarks of TRF1 inhibition (Martinez et al., *Genes & Dev.*, 2009, Sfeir et al., *Cell*, 2009). Moreover, these telomere defects are the same that we previously reported as the consequence of *Trf1* genetic depletion (see Bejarano et al., *Cancer Cell*, 2017; García-Beccaria et al., *EMBO Mol Med*, 2015). In addition, we demonstrate here that kinases of the RAS-MAPK pathway mediate TRF1 phosphorylation (see **Fig. 4**), as well as that non-phosphorylatable mutants in AKT, mTOR and ERK-dependent phosphosites, i.e. T330A, fail to be properly loaded onto telomeres (see **Fig. 5C**, Mendez-Pertuz et al., *Nature Communications*, 2017). Together, these findings point to a direct and causal correlation between TRF1 reduction, TRF1 phosphorylation defects and TRF1 destabilization mediated by the chemical inhibition of specific kinases.

Nevertheless, as requested by the reviewer, in the revised manuscript we now provide further evidence of such functional interaction by genetically depleting ERK1/2 kinase and analyzing TRF1 protein levels as well as telomere dysfunction and telomere damage. In particular, we find that stable RNAi mediated downregulation of ERK1/2 kinases results in a significant decrease in TRF1 protein levels as determined by Western blot (see **new Fig. 6A**) as well as in a significant decrease in the intensity of TRF1 nuclear foci compared to controls with a scrambled oligo (see **new Fig. 6B**).

Moreover, as also suggested by Reviewer #1, we now show in the revised manuscript that telomere damage as measured by TIFs in cells treated with ERK inhibitors, can be rescued by overexpressing a wild-type TRF1-eGFP but not when over-expressing non-phosphorylatable mutant versions of TRF1 (see **new Fig. 5H,I**). To this end, we over-expressed WT TRF1-eGFP and non-phosphorylatable mutant versions of TRF1 in MEFs treated or not with the ERKi (see **new Fig. 5H**). In all cases, treatment with the ERKi decreased the levels of wild-type or mutant TRF1 proteins (see **new Fig. 5H**). Interestingly, the induction of TIFs as the result of ERK1/2 inhibition can be reduced by 2-fold when over-expressing eGFP-Trf1 WT (**new Fig. 5I**) but not when over-expressing the mutant versions of GFP-TRF1 (both single and triple TRF1 mutants in ERK1/2 phosphosites T328, T330 and T335) (**new Fig. 5H,I**), which showed significantly higher amounts of TIFs which were comparable to that observed in control lines treated with ERKi (**new Fig. 5I**). Together, these results indicate that the telomere defects induced by ERKi are related to TRF1 phosphorylation.

Importantly, we confirmed these results observed in the presence of ERK inhibitors by employing a genetic cellular model consisting in MEF stably expressing short hairpin RNAs to silence ERK1/2 (**new Fig. 6A,B**), in which we over-expressed eGFP-tagged versions of TRF1 WT, as well as single and triple TRF1 mutants in T328, T330 and T335 residues (**new Fig. 6C,D**). In particular, we found that TIFs induced by ERK1/2 shRNA were rescued when over-expressing eGFP-Trf1 WT (**new Fig. 6D,E**) but not when over-expressing the mutant versions of GFP-TRF1 (both single and triple TRF1 mutants in ERK1/2 phosphosites T328, T330 and T335) (**new Fig. 6D,E**).

Finally, we would also like to point out that in a parallel project in our group working with the **human TRF1 protein**, we have generated genetic evidence of direct regulation of TRF1 by ERK by using CRISPR-engineered human cell lines *knock-in* for TRF1 in T358 residue (S344 in mouse), which is a site targeted for both AKT- and ERK-mediated phosphorylation (see **new Fig. 4O**; Mendez-Pertuz et al., *Nature Communications*, 2017). These TRF1 *knock-in* cells display

telomere defects attributable to TRF1 loss of function including telomere shortening, increased telomeric damage, and inability to induce telomere elongation upon overexpression of WT Trf1 thus enforcing our evidence pointing to a direct role for ERK in regulating TRF1 function at telomeres (see **Figure 1A,B** for reviewers).

Additional comments:

Figure 2A Does the increased number of TIF stem from a specific telomere effect of the compounds or a general increase in DNA damage, including telomere among others. Thus the authors must normalize their TIF results to the total amount of DNA damage.

ANSWER: As suggested by the reviewer, in the revised manuscript we have now normalized the mean number of TIFs per nucleus in CHA9.3 lung cancer cells to total DNA damage as determined by γ H2AX nuclear intensity (see **new Supplementary Fig. 3C**). The new results clearly indicate that in the majority of the cases the DNA damage stems from telomeres, with the only exception of CDKi and docetaxel that also induce DNA damage elsewhere (see **new Supplementary Fig. 3A**).

Figure 2B; it is the reverse problem than in Figure 2A, an increase in DNA damage is not enough to conclude about any effect on TRF1 or telomere.

ANSWER: That is a good point raised by the reviewer. Glioblastoma stem cells grow in suspension and we had to embed the cells in gelatin for the staining. Therefore, such a feature makes them unsuitable for co-localization of DNA damage with a telomeric marker. For this reason, we performed a γ H2AX immunostaining (**Fig. 2B**) in order to confirm the presence of DNA damage, and then carried out a telomere quantitative FISH (QFISH) experiment (**Fig. 2C**) to address whether part of the DNA damage observed in Figure 2B might be ascribable to specific telomere defects dependent on TRF1 chemical inhibition such as presence of multiple telomere signals or MTSs.

The authors should quote a previous publication relating that TRF2, another shelterin subunit, is targeted by ERK1/2 (Picco et al, Oncotarget, 2016).

ANSWER: We have now quoted that publication in the revised manuscript Discussion (see **page 23, lines 560-565**).

Detailed Answers to Referee #2:

In the current manuscript by Bejarano et al., the authors screened for TRF1 inhibitory drugs using a collection of FDA approved drugs and investigated the effect of these drugs on TRF1 expression. Their experiments revealed that the Ras pathway and PI3K pathway was associated with TRF1 expression by regulation of phosphorylation. In addition, the authors also found that these inhibitors suppressed tumor growth on patient-derived glioma stem cells, especially by a combination of PI3K inhibitors with MEK inhibitors and Gemcitabine.

Considering these results, their findings suggest that TRF1 inhibition may be novel combinatorial therapies in cancer. Although their results are interesting and contribute to the understanding of the roles of TRF1 in cancer biology, some of the key conclusions are not fully supported by the present experiments, as outlined in more detail below. Therefore, I feel that this has to be revised before it can be considered for publication in EMBO Molecular Medicine.

ANSWER: We sincerely appreciated that the Reviewer #2 considers that our “**results are interesting and contribute to the understanding of the roles of TRF1 in cancer biology**”. The reviewer has also a number of insightful suggestions to improve the work, which we have fully addressed in the revised manuscript. In particular:

Major concerns:

1. To test if TRF1 inhibitors (PI3Ki, mTORi, RTKi, ERKi, MEKi, Aurorai, CDKi, PLK1i, Docetaxel, Gemcitabine, and HSP90i) effect on TRF2, TPP1, POT1, TIN2 and RAP1, the authors should show the results of immunoblot analysis of these proteins in CHA-9.3 mouse lung cancer cells treated with TRF1 inhibitors.

ANSWER: We and others have previously published that *Trf1* genetic depletion completely impairs telomere function even in the presence of other shelterin components (Martinez et al., *Genes & Dev.* 2009, Sfeir et al., *Cell*, 2009), thus TRF1 inhibition is sufficient to induce telomere dysfunction. For that reason, we focused our studies specifically on TRF1 genetic and chemical depletion. Nevertheless, in the revised manuscript, we now include a Western Blot showing the levels of TIN2 and RAPI in CHA9.3 mouse lines (see **new Supplementary Fig. 1A,B**). TIN2 was not significantly decreased by any of the compounds found to modulate TRF1. In the case of RAPI, we found a significant decrease by MEK inhibitors (**pages 6-7, lines 157-162**).

2. Page 17, line 417-423: I feel the paragraph has a contradiction. The authors described that the combination of the PI3K inhibitor with Docetaxel resulted toxic in the early stages of the treatment in line 417-419. While the authors described that the combination of the PI3K inhibitor with Docetaxel was well tolerated in the mice in line 419-421. Considering Supplementary Figure 5A, I think the combination of the PI3K inhibitor with Docetaxel has toxic in the mice.

ANSWER: We thank the reviewer for noticing this incongruity, which in fact is a typo: “Among all the possible combinations, the combination of the PI3K inhibitor ETP-47037 with ERKi, RTKi and **HSP90i** resulted toxic in early stages of the treatment”. Although it is true that Docetaxel showed some toxicity (Fig S5A) it occurred in late stages of the treatment. We have now corrected this in the revised version of the manuscript (see **page 19, lines 476-477**).

Minor concerns:

1. Please check all the typo. For example, Page 13, line 330: "TRF1-335" should be corrected with "TRF1-T335"

ANSWER: We have now revised the manuscript text and hope to have corrected all the typos.

Detailed Answers to Referee #3:

General comments:

In this paper, by screening of 114 molecules, the authors identified new molecules to inhibit the telomeric localization of the TRF1 protein (or its expression) and analyzed the implication of such inhibitors (alone or in combination) on tumor growth. The data identified new molecular pathway implicated in TRF1 expression and TRF1 phosphorylation that could impact tumor growth and cancer stemness. More precisely, the author has performed a screen of 114 molecules to study the inhibition of TRF1 expression using Opera screening system. They apply the 114 molecules on CHA-9.3 mouse lung cancer cells line and identified 9 hits that has been secondly tested on same cells by WB. Identified hits are also tested on human Glioma Stem Cells (GSC) with similar effect for most the hits mainly affecting PI3K and Ras pathway and leading to a synergistic effect on tumor growth inhibition and stemness inhibition. This paper can bring interesting novelty in the field and for a more general reader. The introduction clearly place the subject of the article into the general and specific context of the field. The importance of the main data obtained in this paper is also well placed in relation to the field of study and the potential medical consequences. However, there is some limitations in the interpretation and the analysis of the data leading to some overstatement or misleading conclusions (as detailed below). The experiment are technically well done but for some of the figures, the strength of the evidence is not in adequacy with the conclusions drawn as detailed below.

ANSWER: We sincerely appreciate the positive commentaries of Reviewer #3 on the importance and novelty of our work. In particular, the reviewer stated that “**This paper can bring interesting novelty in the field and for a more general reader. The introduction clearly place the subject of the article into the general and specific context of the field. The importance of the main data obtained in this paper is also well placed in relation to the field of study and the potential medical consequences**”. The reviewer has also a number of insightful suggestions to improve the work, which we have fully addressed in the revised manuscript. In particular:

Major concerns:

1. The figure 1 present the results for 9 hits, confirmed on the 2 cell lines in WB and IF. Excepted the Sup Fig 1A showing the Reactome analysis, there is no information of the impact of all the other drugs tested (105 molecules?), even in sup fig or Table. At least a table with the list of the drugs and the quantification of the primary screen should be shown to know why and how the selected the 9 hits.

ANSWER: In the revised manuscript we now add a table with list of drugs emerged from the primary screen as well as the quantification (see **new Supplementary Table 1**).

2. In fig 1C, the DMSO control seems to be loaded with more materials than the other lanes for the first and second panel. Moreover, the duplicate of the third panel is not consistent with less quantity of TRF1 in the right lane compared to the left. The quantity of TRF1 in the right control seems as low as Gem treated cells for instance. How the authors have quantified and normalized the data to compare all the lanes? It is not indicated in method section

ANSWER: In the revised version of the manuscript, we now add information regarding both the quantification and normalization of western blot analysis in the method section (see **page 39, lines 965-968**). In addition, we would like to point out that **Fig. 1C** is a representative image of an experiment repeated at least 3 different times. The variability among the single replicates observed by the reviewer has been taken into account and quantified by using the Standard Error of the Mean (SEM). The bands have been quantified by densitometric analysis with ImageJ software. The Trf1 total levels have been normalized versus SMC1 and the mean of the Trf1/SMC1 ratio deriving from the different replicates have been used to generate the chart (**Fig. 1C**).

3. Since the authors focused their work on glioblastoma for the rest of the paper, why they used the mouse lung cancer cell for the screening?

ANSWER: We appreciate the point raised by the reviewer. In this work we focused on the mouse lung cancer cells (CHA9.3) as well as in GBM. We chose CHA9.3 for the screening as they represent a simpler system for that. Indeed, these cells grow in adhesion and can be plated in Opera plates to perform the high throughput screening. Moreover, they have been already used for the identification of TRF1 chemical inhibitors (Garcia-Beccaria et al., *EMBO Mol Med*, 2015). Thus, this choice allowed us to avoid any additional source of variability. Once identified the new TRF1 chemical inhibitors we validated them in glioblastoma since we have already described it is characterized by high levels of Trf1 which contribute tumor growth (Bejarano et al., *Cancer Cell*, 2017).

4. It appears that CHA93 treatment with the compounds increased the number of cells with TIFs (Fig 2A). It sounds risky to interpret the increase of TIF+ cells after mTOR inhibition "as a trend" as proposed by the authors (line 197-198). To the reviewer point of view, mTOR treatment do not change the proportion of TIF+ cells (15% in control to 18% in mTOR treated). It does not represent a trend to increase. In addition, for TIF counting, why the authors represent the number of cells with at least one TIF? Do they try to count the mean number of TIF per nucleus? In this case, do the mTOR treatment increase DNA damage at telomere? In Fig2B, the authors analyzed the total DNA Damage by quantifying the signal for gH2AX IHC and consider the increase of total DDR as a validation with GSC of the TIF results performed on mouse lung cancer cells. It seems difficult to directly compare these 2 experiments and to use this comparison as a validation. TIF in GSC seems to be more appropriate? Do they try to quantify the signal only for gH2AX in the Fig2A and not only the TIF? In this case, do they find a similar increase of gH2AX they saw in 2B (GSC) after mTOR inhibitor treatment with CHA93?

ANSWER: In the revised version of the manuscript, we state that mTOR inhibitor shows no differences in proportion of cells with $1 < TIFs$ (**page 8, lines 203-204**).

Moreover, as suggested by the reviewer, we have now counted the mean number of TIF per nucleus (see **new Supplementary Fig. 3B**). Again, mTOR treatment does not increase the number of TIFs per nucleus in contrast to ERKi. As already stated above (Answer Reviewer #1), glioblastoma stem cells grow in culture in suspension. Such a feature makes them unsuitable for co-localization of DNA damage with a telomeric marker. For this reason, we first performed a $\gamma H2AX$ immunostaining (see **Fig. 2B**) in order to confirm the presence of DNA damage, and then carried

out a telomere quantitative FISH (QFISH) experiment (see **Fig. 2C**) to address whether part of the DNA damage observed in Figure 2B might be ascribable to specific telomere defects dependent on TRF1 chemical inhibition such as presence of multiple telomere signals or MTSs. Nevertheless, as suggested by reviewers #3 and #1, in the revised manuscript we have now normalized the mean number of TIFs per nucleus in CHA9.3 lung cancer cells to total DNA damage as determined by gH2AX nuclear intensity (see **new Supplementary Fig. 3C**). The new results indicate that in all cases, except for CDKi and docetaxel, the DNA damage steams from telomeres.

5. The response for RTKi seems to be very variable depending the assay. The dose response in 3A for h676 is very flat and RTKi do not seems to have a strong effect, that is consistent with Table 1 where the highest GI50 is obtain for RTKi. However, the effect of RTKi on TRF1 level is the strongest (Fig1C histograms on the right) or AKT phosphorylation (SF2) is the strongest and the effect on TIF (2A) and total gH2AX (2B) is intermediate. Can the authors comment on that?

ANSWER: These are two interesting points raised by the reviewer. However, the two experiments mentioned by the reviewer are completely different thus cannot be compared with each other. Indeed, whereas TRF1 protein levels have been detected following 24h treatment with the drugs (**Fig. 1**), in the dose-response experiments the cells have been incubated for 7 days with the inhibitors without replacing them during the treatment (**Fig 3**). The finding that the RTKi IG50 is low might depend on a possible instability of such inhibitor in long-lasting experiments. We now explain this in the revised manuscript text (**page 10, lines 248-252**).

6. Concerning mouse TRF1 phosphorylation analysis, why the authors performed the experiment using mouse TRF1 with mouse ERK2, mouse MEK1, human bRaf or human mTOR? Why they do not use mouse bRaf and mouse mTOR?

ANSWER: We now clarify in the revised text that

7. The authors claim in the text that the triple mutation T328/T330/T335 is associated with a significant reduction of TRF1 phosphorylation (line 299). Looking at the graph in the figure 4M, we can see a very low decrease but seems to be significant (two stars). Is this effect really significant? At least it seems to be significant but very moderated?

ANSWER: We appreciate this commentary by the reviewer. The decrease in pTRF1 in the triple mutant T328/T330A/T335A was **reproducible and significant** in all the assays we carried out (see **Fig. 4M**). Nevertheless, in the revised version of manuscript we have now performed additional kinase assays including some of the single mutants, such as T328A, T330A and T335A (see **new Fig. 4N**). We have also tested some phosphomutants in residues which were not identified as being ERK-dependent phosphosites, such as the T248 residue whose phosphorylation is mediated by AKT (see **new Fig. 4O**) (Mendez-Pertuz et al., *Nature Communications*, 2017). Interestingly, all the single TRF1 mutants, namely T328A, T330A and T335A showed a statistically significant decrease in their phosphorylated fraction therefore further confirming their role as targets for ERK2-mediated phosphorylation (see **new Fig. 4O**). Furthermore, we found that among the AKT-dependent phosphosites of TRF1, S344 (T358 in human), was as also a target for ERK-mediated phosphorylation. As a negative control, TRF1 mutant in residue T248, which was not identified as an ERK-dependent phosphosite, did not result in decreased TRF1 phosphorylation (see **new Fig. 4O**).

8. In fig5B, endogenous TRF1 is floxed and eGFP TRF1 WT or mutant is re expressed. The authors claimed that endogenous trf1 is deleted (line 310) but the WB do not sustain this conclusion. The 2 controls are not homogenous: the second control is as low as the TRF1 deleted condition or lower (ie 2nd control and T298 or T4S6S7) and TRF1 WT expression is observed all the condition except for the T328T330T335 mutant.

Why all the sample do not look like the T328 T330 T335 mutant? Is the TRF1 deletion complete? Is really a homozygote lox/lox tRF1 mutant? The authors claims in Fig 5C that only the triple T328 T330 T335 mutant is able to inhibit TRF1 telomeric foci: "Quantification of nuclear eGFP spot fluorescence in Trf1^{-/-} MEFs showed that the triple mutant T328/T330/T335 was the only mutant which showed a significant decrease in the intensity of TRF1 telomeric foci compared to MEFs expressing wild-type TRF1 (Fig. 5C)". However, in line with the previous comments, the triple

T328 T330 T335 is the only one showing a total deletion of endogenous TRF1. One interpretation could be that the remaining endogenous TRF1 interfere with the eGFP TRF1 to bring it at telomere. The conclusions of the authors seem overstated. In addition, only the quantification of the eGFP telomeric foci is shown and no images are presented, even in sup. Data should be shown.

ANSWER: We think the reviewer missed some information in Fig 5B. In the western blot, the lane 1 corresponds to the control (not transduced with eGFP-Trf1 or Cre) and the second lane corresponds to the control transduced with Cre, but not with eGFP-TRF1. Thus, in the second lane we show the correct TRF1 excision, which is sufficient to induce a significant decrease in proliferation (see **new Fig. 5D**, purple growth curve). Although we agree that in all the mutants of the Fig 5B we have some signal (with the exception of T328_T330_T335 triple mutant), we attribute it to the fact that these samples have more background in the lanes (see also GFP-TRF1 Panel). Importantly, we would like to point to this reviewer that, all the infections were done in parallel with the same virus. We have now made this more clear in the revised manuscript Methods section to avoid confusions (**page 32, lines 783-788**). Furthermore, in the revised manuscript, we have now repeated the experiment and again show abrogation of the wild-type TRF1 protein by Western blot (see **new Fig. 5E**). In the revised manuscript, we now also show representative images (see **new Supplementary Fig. 5A**).

9. The authors claimed that the triple T328 T330 T335 is the one showing the highest proliferation defect (Fig5E) and claim that it is in agreement with data from fig 5C showing the inhibition of telomeric foci. Again, the data do not sustain this conclusion since the inhibition of proliferation of the triple T328 T330 T335 is very similar to the one of T268 T270 T274 or T4 S6 S7. Is there any statistical difference between these 3 mutants in terms of proliferation?

ANSWER: We thank the reviewer for his/her observation. In the revised version of the manuscript we now plot the data on the same graph and provide the statistical analysis (see **new Supplementary Figure 5B**). The reviewer is right that in his/her statement and we have now re-phrased the conclusions from the proliferation assay as follows:

“Albeit all the mutants showed a partial rescue of the proliferation defects, the presence of specific mutations, such as T330A and T335A, resulted in a significantly lower proliferation rate compared to the wild type GFP-TRF1.” (see **pages 14-15, lines 359-361**).

The T328 T330 T335 mutant is the only one where endogenous TRF1 seems to be fully deleted (see previous comment)

ANSWER: In all the mutants, *Trf1* deletion is similar to our control transduced with Cre (see Fig 5B, second lane). And this condition shows a strong proliferation defect (see Fig 5D, E, F, purple growth curve). Furthermore, in the revised manuscript, we have now repeated the experiment and again show abrogation of the wild-type TRF1 protein by Western blot (see **new Fig. 5E**).

The effect of simple mutant T195 or T298 seems to be similar to the triple. Is the experiment performed in the same time? The black and violet curves is identical in 5D, 5E and 5G suggesting that all the mutant tested are performed in the same experiment. Can the data be plotted on the same graph and statistically analyzed to compare any differences?

ANSWER: All the analysis were performed at the same time. Indeed, the black and violet curves are identical in Fig. 5D,5E and Fig. 5G. The results were represented in 3 different graphs to be visually clearer, but in **new Supplementary Fig 5** we have now plotted all the data in the same graph and statistically compared the differences as mentioned above.

10. Similarly to what they did in Fig 5B-E, the authors want to analyze the effect of single mutation T328, T330 and T335 (Fig5F and G). There is no information available concerning the single mutant T328, T330 and T335 for the expression of endogenous TRF1 like in 5D. Do the endogenous TRF1 is totally deleted? The full interpretation of the effect of single mutant compared to the triple mutant concerning the proliferation or telomeric foci cannot be performed without this WB. In addition, as for Fig5C, images related to Fig5G are missing.

ANSWER: In the revised version of the manuscript, we have now added representative images (see **new Supplementary Fig.5A**) and western blots showing the expression of endogenous as well as the GFP-tagged Trf1 for the single mutants (see **new Fig. 5E**).

11. Concerning data on tumor growth experiment and ETP47037 treatment (Fig S3), the authors said that the treatment start one week after GSC injection. DO the tumor are already palpable? DO they try to treat after the establishment of the tumor and not before? This could change the interpretation of the data.

ANSWER: The tumors were palpable and already established when we started the treatment. We now include this information in the Materials and Methods section of the revised manuscript.

12. The authors measured TRF1 expression level at the end of the tumor growth experiment and they do not see any difference between treated and not treated, likely due to the re expression of TRF1. But their conclusion is overstated. S3D and S3E do not "indicate" but "suggest" that the regrowth is associated to TRF1 re expression (line 362).

ANSWER: We appreciate this commentary by the reviewer. In the revised version of the manuscript we have rephrased this sentence as suggested by the reviewer (**page 17, line 420**).

Do they try to compare the TRF1 expression level in tumors treated and untreated at days 20 or day 30 when there is a strong difference of growth? In this case, do TRF1 is really less expressed? It could be also because the treatment select the cells with the highest expression of TRF1

ANSWER: For the reviewer information, we already reported previously that the xenografts derived from h676 and h543 GCSs showed a significant reduction in Trf1 levels upon ETP-47030 treatment compared to the untreated controls at 20 and 30 days post-injection (Figures 8C, 8D and 8F in Bejarano et al, *Cancer Cell*, 2017).

13. The data on neurospheres are convincing (6A-F and sup 4C-H). The quantification of neurospheres could be more appreciated if presented in box plot with all point as the authors did for Sup Fig 4 for neurosphere diameter. Concerning WB for TRF1 after treatment, the quantification is wired on some aspect. For instance, the blot in 6H give the feeling that comb treatment is really synergistic and induce a strong decrease compared to control but the quantification does not sustain this decrease. Moreover, axis is not labelled. On the other hand, the quantification of the comb treatment in 6J is similar to the one in 6H but the blot signal seems to be higher. Are the authors be sure of the quantification? The decrease of TRF1, except for 6G and 6I is not synergistic while the effect on neurosphere seems to be synergistic. How the author can conciliate this discrepancy?

ANSWER: We appreciate reviewer's advice on neurosphere data representation and in the revised version of the manuscript we have now plotted the number of neurospheres as a box-plot (see **new Figure 7A-F**) Also, regarding the quantification, we would like to point out that the blots only represent 1 of all the samples we checked, while the graphs include all the samples as indicated by the n. Regarding the Western blot for TRF1 after treatment, we would like to point out that the figures 6H and 6J are representative images of experiments repeated at least 5 different times. Thus, as the charts derive from the mean value of at least 5 replicates, we do not expect they perfectly correlate with the image from a single WB, especially if we consider the intrinsic variability of each replicate. Nevertheless, the apparent discrepancy among the representative images and the charts observed by the reviewer has been taken into account and quantified by using the Standard Error of the Mean (SEM). Regarding the discrepancy between the effect on Trf1 levels and the effect in neurosphere formation, as we already explained in the answer #5 to this reviewer, we cannot correlate the synergistic effect of drug combinations in neurosphere growth with TRF1 levels as the two above mentioned experiments are completely different. Indeed, whereas TRF1 levels have been detected following 24h treatment with the drugs, in the neurosphere experiments the cells have been incubated for 7 days with the inhibitors without replacing them during the treatment.

14. The effect of comb. Treatment on tumor growth is clear and appears synergistic (Fig7A-D and SF5). The author do not precise the number of mice per condition and should indicate it. It is not clear for the comb treatment that give toxicity how the author manage to perform the experiment: if

100% of the mice are dying for toxicity as shown in Sup F5A for PI3Ki + RTKi, how they succeed to obtain the data in 7A ?

ANSWER: As suggested by the reviewer, in the revised manuscript we now indicate the number of mice per condition (see **new Figure 8A-D**). Regarding the toxic combinations, the curves have been plotted until the last mouse died, that is how we obtained the data in Figure 7A. We now include this explanation in the revised Methods section (**page 37, lines 930-931**).

15. In vitro and in vivo data are opposite in some cases like gemcitabine treatment. While gemcitabine alone or in comb with PI3K do not decrease TRF1 expression level in tumor sample, gemcitabine decrease TRF1 expression (Fig1), increase TIF and total gH2AX (fig 2), decrease number and diameter of neurospheres (fig3 and SF4), in synergy with PI3K inhibitor (fig 6E and 6K) and synergistically decrease tumor growth (fig 7C). How the author can explain this discrepancy?

ANSWER: We agree with the reviewer. In some cases we do not see *in vivo* the same effects as *in vitro*. In the case of gemcitabine, probably the *in vivo* synergistic effect is due to a Trf1-independent mechanism. We have now discussed this in the revised version of the manuscript (**page 24, lines 577-579**).

Minor concerns:

16. It is not clear why the authors perform initially their screen on a mouse lung cancer cell line and then perform all the rest of the study on GSC using either H676 cells or H543 or both.

ANSWER: We already answered this question in the point 3 of the same reviewer.

17. The authors claim that they demonstrate "an unprecedented role of the Ras pathway in telomere protection" (line 129-130 and 493). However, it has already been shown that Ras pathway could be implicated in telomere protection since it has been shown that Ras overexpression protect cancer cells from DDR induced by TRF2 dysfunction through a IL-6 dependent mechanism (Biroccio et al 2013 NCB). It could be interesting to confront these data and to include it in the discussion.

ANSWER: We have included this in the discussion in the revised version of the manuscript (see **page 23, lines 565-567**).

18. The legend of the fig1 is not complete and the annotation of the panel of the Fig 1 is misleading. What are the cells used for the quantification of TRF1 in the 2 histograms of the Fig1 just above the scheme of signaling pathway? Is it H676?

ANSWER: The cells used in the Fig1A are CHA9.3 mouse lung cancer lines, while the cells in Figure 1B are h676. We have now clarified this in the legend of **Figure 1** in the revised version of the manuscript.

19. All over the text, authors refer to a "STAR method" like line 159 that is not formally present in the manuscript.

ANSWER: We now included all Methods in the Materials and Methods section.

20. In Fig 3 and Table 1, the authors determined using the H676 and H543 GSC whether the treatment with the inhibitors alter the stemness of human GSC since they showed previously that TRF1 OE in GSC increase stemness. But, there is some unclear information: TRF1 inhibition decrease stemness in the 2 cell lines tested. why the authors do not perform the DDR analysis on H543 cells? Then, in the Table 1, there is probably a typo error. Is it really h676 or h767 cells used?

ANSWER: As stated by this reviewer, we showed that the tested compounds inhibit the stemness of both h676 and h543 GSC. Thus, as the effect of such inhibitors was exactly the same on both the GSC lines, likewise we reasoned that their effect on DNA damage might be the same, thus we simply chose only one among those lines to perform the DDR analysis.

“there is probably a typo error”.

ANSWER: Yes, it is h676. We have now corrected typos and edition problems in the revised version of the manuscript.

21. Various typo problems or edition problems (line 327; line 230; star methods everywhere...)

ANSWER: We have now corrected typos and edition problems in the revised version of the manuscript.

22. Line 427: do the authors really refers to fig S4D and E?

ANSWER: Yes, this was a mistake. We have replaced Fig. S4D-E with Figure 7E-H in the revised version of the manuscript

REFERENCES:

Sfeir A et al. (2009) ‘Mammalian telomeres resemble fragile sites and require TRF1 for efficient replication’, *Cell*, 138(1), pp.90-103, doi:10.1016/j.cell.2009.06.021

Martinez P et al. (2009) ‘Increased telomere fragility and fusions resulting from TRF1 deficiency lead to degenerative pathologies and increased cancer in mice’, *Genes and Development*, 23(17), pp. 2060-2’75. doi: 10.1101/gad.543509

Méndez-Pertuz, M. et al. (2017) ‘Modulation of telomere protection by the PI3K/AKT pathway’, *Nature Communications*, 8(1). doi: 10.1038/s41467-017-01329-2.

Garcia-Beccaria, M. et al. (2015) ‘Therapeutic inhibition of TRF1 impairs the growth of p53-deficient K-RasG12V-induced lung cancer by induction of telomeric DNA damage.’ *EMBO Mol. Med.* 7, 930–949, doi: 10.15252/emmm.201404497.

Bejarano, L. et al. (2017) ‘Inhibition of TRF1 Telomere Protein Impairs Tumor Initiation and Progression in Glioblastoma Mouse Models and Patient-Derived Xenografts’, *Cancer Cell*, 32(5), p. 590–607.e4. doi: 10.1016/j.ccell.2017.10.006.

2nd Editorial Decision

3 May 2019

Thank you for the submission of your revised manuscript to EMBO Molecular Medicine. We have received the enclosed reports from the referees, who are now supportive of publication. I am therefore pleased to inform you that we will be able to accept your manuscript once the following final editorial amendments have been completed.

***** Reviewer's comments *****

Referee #1 (Remarks for Author):

The revised version adequately addressed my points

Referee #2 (Remarks for Author):

Now this paper is suitable for publication in EMBO Molecular Medicine.

Referee #3 (Remarks for Author):

The authors answered to almost all points raised after the review and greatly enhance the manuscript. Under its current form, I think that this paper can be published in EMBO Mol Med.

2nd Revision - authors' response

10 May 2019

Authors made the requested editorial changes.

Corresponding Author Name: Maria A. Blasco
 Journal Submitted to: EMBO molecular medicine
 Manuscript Number: EMM-2019-10292-V2